# A neurotransmitter atlas of *C. elegans* males and hermaphrodites

Chen Wang[1]*, Berta Vidal[1], Surojit Sural[1], Curtis Loer[2], G Robert Aguilar[1], Daniel M Merritt[1], Itai Antoine Toker[1], Merly C Vogt[1†], Cyril C Cros[1‡], Oliver Hobert[1]*

[1]Department of Biological Sciences, Howard Hughes Medical Institute, Columbia University, New York, United States; [2]Department of Biology, University of San Diego, San Diego, United States

*For correspondence:
cw2955@columbia.edu (CW);
or38@columbia.edu (OH)

Present address: †Institute for Diabetes and Cancer, Helmholtz Center, Munich, Germany; ‡European Molecular Biology Institute, Heidelberg, Germany

Competing interest: The authors declare that no competing interests exist.

**Abstract** Mapping neurotransmitter identities to neurons is key to understanding information flow in a nervous system. It also provides valuable entry points for studying the development and plasticity of neuronal identity features. In the *Caenorhabditis elegans* nervous system, neurotransmitter identities have been largely assigned by expression pattern analysis of neurotransmitter pathway genes that encode neurotransmitter biosynthetic enzymes or transporters. However, many of these assignments have relied on multicopy reporter transgenes that may lack relevant *cis*-regulatory information and therefore may not provide an accurate picture of neurotransmitter usage. We analyzed the expression patterns of 16 CRISPR/Cas9-engineered knock-in reporter strains for all main types of neurotransmitters in *C. elegans* (glutamate, acetylcholine, GABA, serotonin, dopamine, tyramine, and octopamine) in both the hermaphrodite and the male. Our analysis reveals novel sites of expression of these neurotransmitter systems within both neurons and glia, as well as non-neural cells, most notably in gonadal cells. The resulting expression atlas defines neurons that may be exclusively neuropeptidergic, substantially expands the repertoire of neurons capable of co-transmitting multiple neurotransmitters, and identifies novel sites of monoaminergic neurotransmitter uptake. Furthermore, we also observed unusual co-expression patterns of monoaminergic synthesis pathway genes, suggesting the existence of novel monoaminergic transmitters. Our analysis results in what constitutes the most extensive whole-animal-wide map of neurotransmitter usage to date, paving the way for a better understanding of neuronal communication and neuronal identity specification in *C. elegans*.

## eLife assessment

This **fundamental** study reports the most comprehensive neurotransmitter atlas of any organism to date, using fluorescent knock-in reporter lines. The work is comprehensive, rigorous, and **compelling**. The tool will be used by broad audience of scientists interested in neuronal cell type differentiation and function, and could be a seminal reference in the field.

## Introduction

Understanding information processing in the brain necessitates the generation of precise maps of neurotransmitter deployment. Moreover, comprehending synaptic wiring diagrams is contingent upon decoding the nature of signaling events between anatomically connected neurons. Mapping of neurotransmitter identities onto individual neuron classes also presents a valuable entry point for studying how neuronal identity features become genetically specified during development and

potentially modified in response to specific external factors (such as the environment) or internal factors (such as sexual identity or neuronal activity patterns).

The existence of complete synaptic wiring diagrams of the compact nervous system of male and hermaphrodite *Caenorhabditis elegans* nematodes raises questions about the molecular mechanisms by which individual neurons communicate with each other. *C. elegans* employs the main neurotransmitter systems that are used throughout the animal kingdom, including acetylcholine, glutamate, γ-aminobutyric acid (GABA), and several monoamines (*Sulston et al., 1975*; *Horvitz et al., 1982*; *Loer and Kenyon, 1993*; *McIntire et al., 1993*; *Duerr et al., 1999*; *Lee et al., 1999*; *Duerr et al., 2001*; *Alkema et al., 2005*; *Duerr et al., 2008*; *Serrano-Saiz et al., 2013*; *Pereira et al., 2015*; *Gendrel et al., 2016*; *Serrano-Saiz et al., 2017b*; *Figure 1A*). Efforts to map these neurotransmitter systems to individual cell types throughout the entire nervous system have a long history, beginning with the use of chemical stains that directly detected a given neurotransmitter (dopamine) (*Sulston et al., 1975*), followed by antibody staining of neurotransmitter themselves (serotonin and GABA) (*Horvitz et al., 1982*; *McIntire et al., 1993*) or antibody stains of biosynthetic enzymes or neurotransmitter vesicular transporters (acetylcholine and monoamines) (*Loer and Kenyon, 1993*; *Duerr et al., 1999*; *Duerr et al., 2001*; *Alkema et al., 2005*; *Duerr et al., 2008*; see *Figure 1A* for an overview of these enzymes and transporters).

While these early approaches proved successful in revealing neurotransmitter identities, they displayed several technical limitations. Since neurotransmitter-synthesizing or -transporting proteins primarily localize to neurites, the cellular identity of expressing cells (usually determined by assessing cell body position) often could not be unambiguously established in several, particularly cell- and neurite-dense regions of the nervous system. One example concerns cholinergic neurons, which are defined by the expression of the vesicular acetylcholine transporter UNC-17/VAChT and choline acetyltransferase CHA-1/ChAT. While mainly neurite-localized UNC-17 and CHA-1 antibody staining experiments could identify a subset of cholinergic neurons (*Duerr et al., 2001*; *Duerr et al., 2008*), many remained unidentified (*Pereira et al., 2015*). In addition, for GABA-producing neurons, it became apparent that antibody-based GABA detection was dependent on staining protocols, leading to the identification of 'novel' anti-GABA-positive neurons, i.e., GABAergic neurons, more than 20 years after the initial description of GABAergic neurons (*McIntire et al., 1993*; *Gendrel et al., 2016*).

An alternative approach to mapping neurotransmitter usage has been the use of reporter transgenes. This approach has the significant advantage of allowing the fluorophore to either fill the entire cytoplasm of a cell or to be targeted to the nucleus, thereby facilitating neuron identification. However, one shortcoming of transgene-based reporter approaches is that one cannot be certain that a chosen genomic region, fused to a reporter gene, indeed contains all *cis*-regulatory elements of the respective locus. In fact, the first report that described the expression of the vesicular glutamate transporter EAT-4, the key marker for glutamatergic neuron identity, largely underestimated the number of *eat-4/VLGUT*-positive and, hence, glutamatergic neurons (*Lee et al., 1999*). The introduction of fosmid-based reporter transgenes has largely addressed such concerns, as these reporters, with their 30–50 Kb size, usually cover entire intergenic regions (*Sarov et al., 2012*). Indeed, such fosmid-based reporters have been instrumental in describing the supposedly complete *C. elegans* glutamatergic nervous system, defined by the expression of *eat-4/VGLUT* (*Serrano-Saiz et al., 2013*), as well as the supposedly complete set of cholinergic (*Pereira et al., 2015*) and GABAergic neurons (*Gendrel et al., 2016*).

However, even fosmid-based reporters may not be the final word. In theory, they may still miss distal *cis*-regulatory elements. Moreover, the multicopy nature of transgenes harbors the risk of overexpression artifacts, such as the titrating of rate-limiting negative regulatory mechanisms. Also, RNAi-based silencing mechanisms triggered by the multicopy nature of transgenic reporter arrays have the potential to dampen the expression of reporter arrays (*Nance and Frokjaer-Jensen, 2019*). One way to get around these limitations, while still preserving the advantages of reporter gene approaches, is to generate reporter alleles in which an endogenous locus is tagged with a reporter cassette, using CRISPR/Cas9 genome engineering. Side-by-side comparisons of fosmid-based reporter expression patterns with those of knock-in reporter alleles indeed revealed several instances of discrepancies in expression patterns of homeobox genes (*Reilly et al., 2022*).

An indication that previous neurotransmitter assignments may not have been complete was provided by recent single-cell RNA (scRNA) transcriptomic analyses of the hermaphrodite nervous

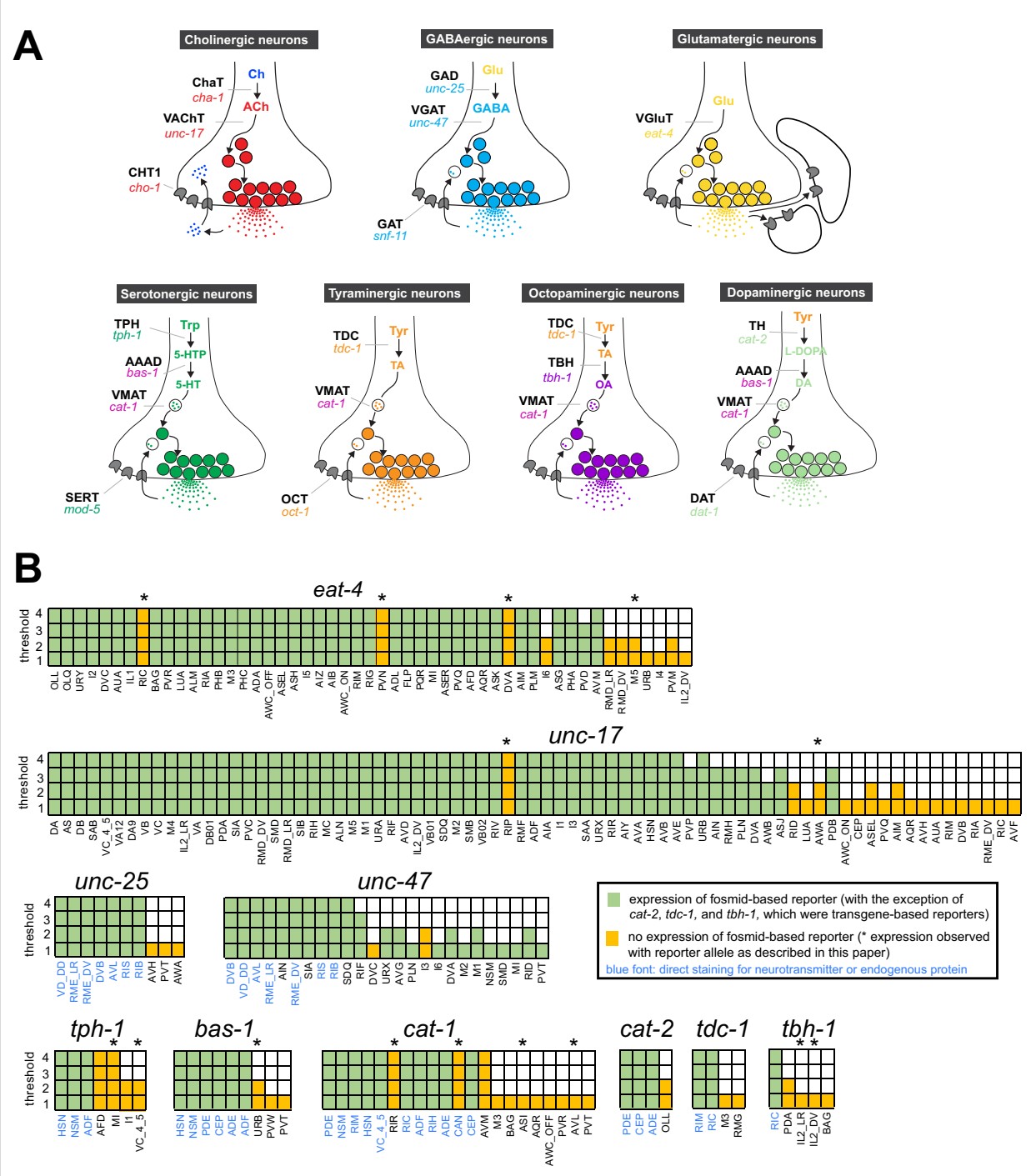

**Figure 1.** Background on genes examined in this paper. (**A**) Neurotransmitter synthesis and transport pathways. TH = tyrosine hydroxylase; TDC = tyrosine decarboxylase; TBH = tyramine β-hydroxylase; TPH = tryptophan hydroxylase; GAD = glutamic acid decarboxylase; AAAD = aromatic amino acid decarboxylase; VMAT = vesicular monoamine transporter; VAChT = vesicular acetylcholine transporter; VGAT = vesicular γ-aminobutyric acid (GABA) transporter; Ch = choline; ACh = acetylcholine; TA = tyramine; OA = octopamine; DA = dopamine. CHT1 = choline uptake transporter; SERT = serotonin uptake transporter; OCT = organic cation transporter; DAT = dopamine uptake transporter; GAT = GABA uptake transporter. Taken and modified from Figure 6 of *Hobert, 2013*. (**B**) Graphic comparison of single-cell RNA (scRNA) expression data and previously reported reporter expression data. See *Supplementary file 1* for a more comprehensive version that includes expression of reporter genes in cells that show no scRNA transcripts. Note that scRNA expression values for *eat-4* and *unc-47* can be unreliable because they were overexpressed to isolate individual neurons for scRNA analysis (*Taylor et al., 2021*).

The online version of this article includes the following figure supplement(s) for figure 1:

**Figure supplement 1.** Use of aromatic amino acid decarboxylases (AAADs) in *C. elegans*.

system of L4 stage animals by the CeNGEN consortium (*Taylor et al., 2021*). As we describe in this paper in more detail, transcripts for several neurotransmitter-synthesizing enzymes or transporters were detected in a few cells beyond those previously described to express the respective reporter genes. This motivated us to use CRISPR/Cas9 engineering to fluorescently tag a comprehensive panel of genetic loci that code for neurotransmitter-synthesizing, -transporting, and -uptaking proteins ('neurotransmitter pathway genes'). Using the landmark strain NeuroPAL for neuron identification (*Yemini et al., 2021*), we identified novel sites of expression of most neurotransmitter pathway genes. Furthermore, we used these reagents to expand and refine neurotransmitter maps of the entire nervous system of the *C. elegans* male, which contains almost 30% more neurons than the nervous system of the hermaphrodite yet lacks a reported scRNA transcriptome atlas. Together with the NeuroPAL cell-identification tool, these reporter alleles allowed us to substantially improve the previously described neurotransmitter map of the male nervous system (*Serrano-Saiz et al., 2017b*). Our analysis provides insights into the breadth of usage of each individual neurotransmitter system, reveals instances of co-transmitter use, indicates the existence of neurons that may entirely rely on neuropeptides instead of classic neurotransmitters, reveals sexual dimorphisms in neurotransmitter usage, and suggests the likely existence of presently unknown neurotransmitters.

## Results
### Comparing CeNGEN scRNA data to reporter gene data

To investigate the neurotransmitter identity of neurons throughout the entire *C. elegans* nervous system of both sexes, we consider here the expression pattern of the following 15 genetic loci (see also *Figure 1A*):

a. *eat-4/VGLUT*: expression of the vesicular glutamate transporter is alone sufficient to define glutamatergic neuron identity (*Lee et al., 1999*; *Serrano-Saiz et al., 2013*).

b. *unc-17/VAChT*: expression of the vesicular acetylcholine transporter, located in an operon together with the acetylcholine-synthesizing gene *cha-1/ChAT* (*Alfonso et al., 1994*), defines cholinergic neurons (*Duerr et al., 2001*; *Duerr et al., 2008*; *Pereira et al., 2015*).

c. *unc-25/GAD, unc-47/VGAT,* and its sorting co-factor *unc-46/LAMP*: expression of these three genes defines neurons that synthesize and release GABA (*McIntire et al., 1993*; *McIntire et al., 1997*; *Jin et al., 1999*; *Schuske et al., 2007*; *Gendrel et al., 2016*). Additional neurons that we classify as GABAergic are those that do not synthesize GABA (*unc-25/GAD*-negative), but take up GABA from other neurons (based on anti-GABA antibody staining) and are expected to release GABA based on *unc-47/VGAT* expression (*Gendrel et al., 2016*). *unc-47/VGAT* expression without any evidence of GABA synthesis or uptake (*unc-25/GAD*- and anti-GABA-negative) is indicative of an unknown transmitter being present in these cells and utilizing *unc-47/VGAT* for vesicular secretion.

d. *tph-1/TPH* and *bas-1/AAAD*: the co-expression of these two biosynthetic enzymes, together with the co-expression of the monoamine vesicular transporter *cat-1/VMAT*, defines all serotonin-synthesizing and -releasing neurons (*Figure 1A*; *Horvitz et al., 1982*; *Duerr et al., 1999*; *Sze et al., 2000*; *Hare and Loer, 2004*).

e. *cat-2/TH* and *bas-1/AAAD*: the co-expression of these two biosynthetic enzymes, together with the co-expression of the monoamine vesicular transporter *cat-1/VMAT*, defines all dopamine-synthesizing and -releasing neurons (*Figure 1A*; *Sulston et al., 1975*; *Duerr et al., 1999*; *Lints and Emmons, 1999*; *Hare and Loer, 2004*).

f. *tdc-1/TDC*: defines, together with *cat-1/VMAT,* all tyramine-synthesizing and -releasing neurons (*Figure 1A*; *Alkema et al., 2005*).

g. *tbh-1/TBH*: expression of this gene, in combination with that of *tdc-1/TDC* and *cat-1/VMAT*, defines octopamine-synthesizing and -releasing neurons (*Figure 1A*; *Alkema et al., 2005*).

h. *cat-1/VMAT*: expression of this vesicular monoamine transporter defines all four above-mentioned monoaminergic neurons (serotonin, dopamine, tyramine, octopamine) (*Duerr et al., 1999*), but as described and discussed below, it may also define additional sets of monoaminergic neurons.

i. *hdl-1/AAAD: hdl-1,* a previously uncharacterized gene, encodes the only other AAAD with sequence similarity to the *bas-1* and *tdc-1* AAAD enzymes that produce other bona fide

monoamines (*Figure 1—figure supplement 1*; *Hare and Loer, 2004*). *hdl-1* expression may therefore, in combination with *cat-1/VMAT*, identify neurons that produce and release trace amines of unknown identity.

j. *snf-3/BGT1/SLC6A12*: this gene encodes the functionally validated ortholog of the vertebrate betaine uptake transporter SLC6A12 (i.e. BGT1) (*Peden et al., 2013*). In combination with the expression of *cat-1/VMAT*, which synaptically transports betaine (*Hardege et al., 2022*), *snf-3* expression may identify neurons that utilize betaine as a synaptically released neurotransmitter to gate betaine-gated ion channels, such as ACR-23 (*Peden et al., 2013*) or LGC-41 (*Hardege et al., 2022*).

k. *mod-5/SERT*: this gene codes for the functionally validated ortholog of the vertebrate serotonin uptake transporter SERT (*Ranganathan et al., 2001*), which defines neurons that take up serotonin independently of their ability to synthesize serotonin and, depending on their expression of *cat-1/VMAT*, may either re-utilize serotonin for synaptic signaling or serve as serotonin clearance neurons.

l. *oct-1/OCT*: this gene encodes the closest representative of the OCT subclass of SLC22 organic cation transporters (*Zhu et al., 2015*), several members of which are selective uptake transporters of tyramine (*Breidert et al., 1998*; *Berry et al., 2016*). Its expression or function in the nervous system had not previously been analyzed in *C. elegans*.

For all these 15 genetic loci, we compared scRNA transcriptome data from the CeNGEN scRNA atlas (at all four available stringency levels; *Taylor et al., 2021*) to previously published reporter and antibody staining data. As shown in *Figure 1B* and *Supplementary file 1*, such comparisons reveal the following: (a) scRNA data support the expression of genes in the vast majority of neurons in which those genes were found to be expressed with previous reporter gene approaches. In most cases, this is true even at the highest threshold levels for scRNA detection. (b) Vice versa, reporter gene expression supports scRNA transcriptome data for a specific neurotransmitter system in the great majority of cells. (c) In spite of this congruence, there were several discrepancies between reporter data and scRNA data. Generally, while valuable, scRNA transcriptome data cannot be considered the final word for any gene expression pattern assignments. Lack of detection of transcripts could be a sensitivity issue and, conversely, the presence of transcripts does not necessarily indicate that the respective protein is generated, due to the possibility of posttranscriptional regulation.

Hence, to consolidate and further improve neurotransmitter identity assignment throughout the entire *C. elegans* nervous system, and to circumvent potential limitations of multicopy, fosmid-based reporter transgenes on which previous neurotransmitter assignments have been based, we engineered and examined expression patterns of 16 knock-in reporter alleles of the 15 neurotransmitter synthesis, vesicular transport, and uptake loci listed above (*Figure 1*, *Figure 2*). For *unc-17* and *eat-4*, we knocked-in a *t2a::gfp::h2b* (*his-44*) cassette right before the stop codon of the respective gene. For *unc-25*, we created two knock-in alleles with the *t2a::gfp::h2b* (*his-44*) cassette tagging isoforms a.1/c.1 and b.1 separately. For *tdc-1*, a *gfp::h2b::t2a* cassette was knocked into the N-terminus of the locus because of different C-terminal splice variants. The self-cleaving T2A peptide frees up GFP::H2B, which will be transported to the nucleus, thereby facilitating cell identification. For *unc-46*, *unc-47*, *tph-1*, *bas-1*, *tbh-1*, *cat-1*, *cat-2*, *snf-3*, and *oct-1*, we knocked-in a *sl2::gfp::h2b* cassette at the C-terminus of the locus. The SL2 sequence also provides for the separate production of GFP::H2B. Both types of reporter cassettes should capture posttranscriptional, 3'UTR-mediated regulation of each locus, e.g., by miRNAs and RNA-binding proteins (not captured by CeNGEN scRNA data). Since in each case the reporter is targeted to the nucleus, this strategy circumvents shortcomings associated with interpreting antibody staining patterns or dealing with too densely packed cytosolic signals. For *mod-5*, we analyzed a previously generated, non-nuclear reporter allele (*Maicas et al., 2021*). For all our neuronal cell identification, we utilized the neuronal landmark strain NeuroPAL (*Tekieli et al., 2021*; *Yemini et al., 2021*). The results of our neuronal expression pattern analysis are summarized in *Figure 3* and detailed in *Supplementary files 2 and 3*. In the ensuing sections we describe these patterns in detail.

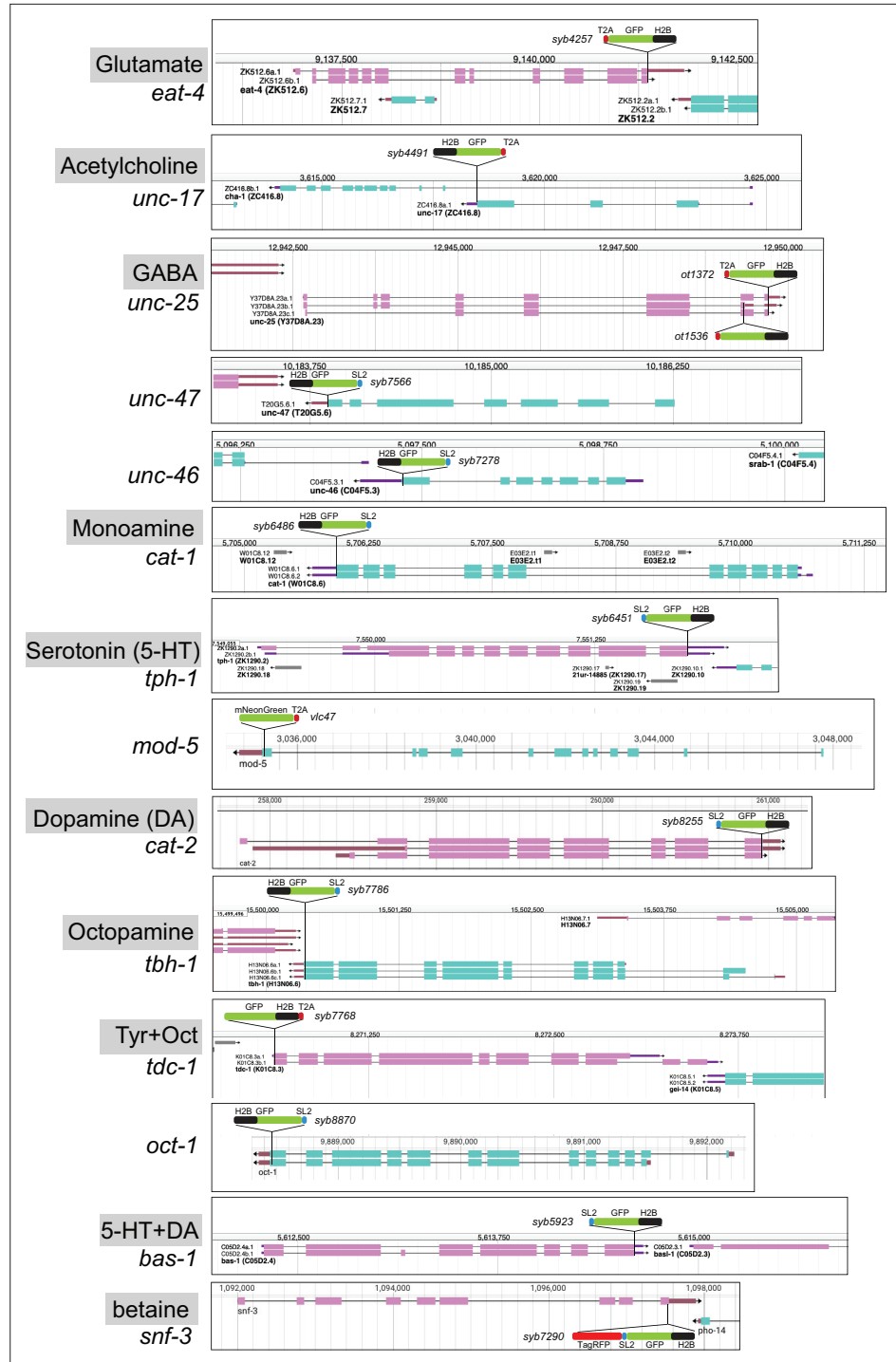

**Figure 2.** Schematics of reporter knock-in alleles. Reporter alleles were generated by CRISPR/Cas9 genome engineering. The SL2- or T2A-based separation of the reporter from the coding sequence of the respective loci enables targeting of the reporter to the nucleus (via the H2B tag), which in turn facilitates the identification of the cell expressing a given reporter. Genome schematics are from WormBase (*Davis et al., 2022*). See *Figure 1—figure supplement 1* for *hdl-1* reporter alleles.

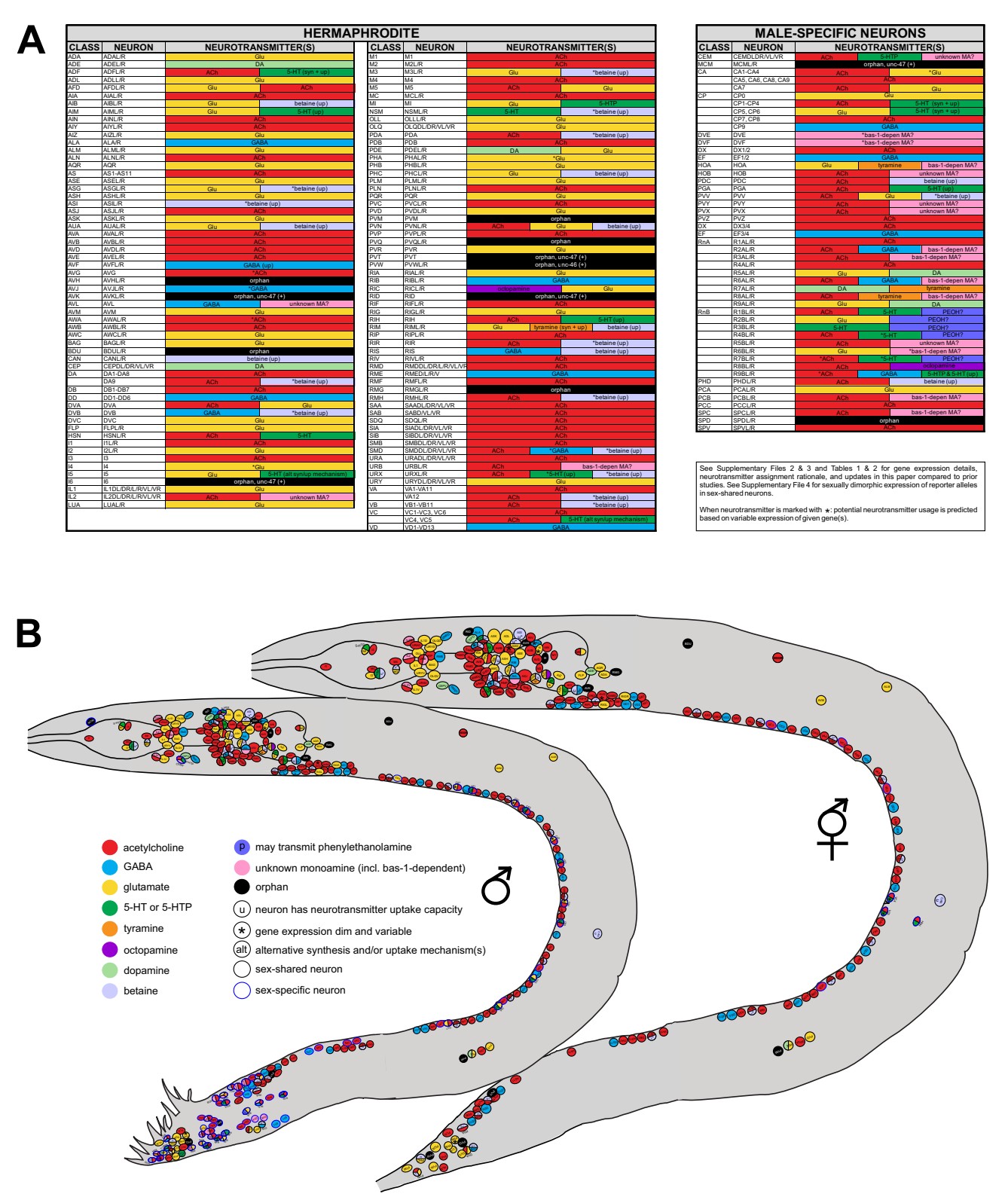

**Figure 3.** Summary of neurotransmitter usage and atlases. See *Table 1*, *Table 2*, and *Supplementary files 2–4* for individual gene expression, rationale for neurotransmitter assignments, and more detailed notes. (**A**) ACh=acetylcholine; Glu=glutamate; GABA=γ-aminobutyric acid; DA=dopamine; 5-HT=5-hydroxytryptamine, or serotonin; 5-HTP=5-hydroxytryptophan; PEOH?=the neuron has the potential to use β-hydroxyphenethylamine, or phenylethanolamine; *bas-1*-depen MA?=the neuron has the potential to use *bas-1*-dependent unknown monoamines (histamine, tryptamine,

*Figure 3 continued on next page*

*Figure 3 continued*

phenylethylamine [PEA]; also see *Figure 1—figure supplement 1*); unknown MA?=the neuron has the potential to use non-canonical monoamines; (up)=neurotransmitter uptake; (syn)=neurotransmitter synthesis; *=dim and variable expression of respective identity gene(s) is detected. Variability could be due to one of the following reasons: (1) the endogenous gene is indeed expressed in some but not all animals; (2) the endogenous gene is indeed expressed in every animal but the level of reporter expression is below detection threshold in some. Variability is detected only at low fluorescent intensity; at higher intensities, expression remains consistent. Results for anti-γ-aminobutyric acid (GABA) staining in SMD and anti-serotonin staining in VC4, VC5, CEM, I5, and URX are variable based on previous reports (see text for citations). (B) Information from (A) shown in the context of neuron positions in worm schematics. Note 'unknown monoamine' here includes both '*bas-1*-depen MA' and 'unknown MA' in (A). Neurons marked with 'u' can uptake given neurotransmitters but not exclusively; some may also synthesize them, e.g., ADF can both synthesize and uptake serotonin.

## Expression of a reporter allele of *eat-4/VGLUT*, a marker for glutamatergic identity, in the hermaphrodite

37 of the 38 previously reported neuron classes that express an *eat-4* fosmid-based reporter (*Serrano-Saiz et al., 2013*) showed *eat-4* transcripts in the CeNGEN scRNA atlas (*Taylor et al., 2021*) at all four thresholds of stringency, and 1/38 (PVD neuron) showed it in three out of the four threshold levels (*Figure 1B*, *Supplementary file 1*). However, scRNA transcripts were detected at all four threshold levels in three additional neuron classes, RIC, PVN, and DVA, for which no previous reporter data provided support. In a recent publication, we had already described that the *eat-4* reporter allele *syb4257* is expressed in RIC (*Reilly et al., 2022*) (confirmed in *Figure 4A*). We now also confirm expression of this reporter allele, albeit at low levels, in DVA and PVN (*Figure 4B*, *Supplementary file 2*).

Another neuron found to have some *eat-4* transcripts, but only with the two lower threshold sets, is the I6 pharyngeal neuron. Consistent with our previous fosmid-based reporter data, we detected no I6 expression with our *eat-4(syb4257)* reporter allele. The *eat-4* reporter allele also shows expression in the pharyngeal neuron M5, albeit very weakly (*Figure 4A*, *Supplementary file 2*), consistent with CeNGEN scRNA data. Weak expression of the *eat-4* fosmid-based reporter in ASK and ADL remained weak, but clearly detectable with the *eat-4(syb4257)* reporter allele (*Figure 4A*, *Supplementary file 2*). Extremely dim expression in PHA can be occasionally detected. Whereas the PVQ neuron class displays *eat-4* scRNA transcripts and was reported to show very dim *eat-4* fosmid-based reporter expression, we detected no expression of the *eat-4(syb4257)* reporter allele in PVQ neurons (*Figure 4B*, *Supplementary file 2*). We also did not detect expression of *eat-4(syb4257)* in the GABAergic AVL and DVB neurons, in which a recent report describes expression of an *eat-4* promoter fusion reporter (*Li et al., 2023*). An absence of *eat-4(syb4257)* expression in AVL and DVB is also consistent with the absence of scRNA transcripts in these neurons.

A few neurons were found to express *eat-4* transcripts by the CeNGEN atlas, but only with lower threshold levels, including, for example, the RMD, PVM, and I4 neurons (*Figure 1B*, *Supplementary file 1*). We failed to detect reporter allele expression in RMD or PVM neurons, but occasionally observed very dim expression in I4. Lastly, we identified a novel site of *eat-4* expression in the dopaminergic PDE neuron (*Figure 4B*, *Supplementary file 2*). While such expression was neither detected with previous reporters nor scRNA transcripts, we detected it very consistently but at relatively low levels.

## Expression of a reporter allele of *unc-17/VAChT*, a marker for cholinergic identity, in the hermaphrodite

41 of previously described 52 neuron classes that show *unc-17* fosmid-based reporter expression (*Pereira et al., 2015*) showed transcripts in the CeNGEN scRNA atlas at four out of four threshold levels, another seven neuron classes at three out of four threshold levels, and one at the lowest two threshold levels (*Taylor et al., 2021*). Only one neuron class, RIP, displayed scRNA levels at all four thresholds, but showed no corresponding *unc-17* fosmid-based reporter expression (*Figure 1B*, *Supplementary file 1*). Using the *unc-17(syb4491)* reporter allele (*Figure 1A*), we confirmed expression in RIP (*Figure 4C*, *Supplementary file 2*). Of the additional neuron classes that show *unc-17* expression at the lower stringency transcript detection levels (*Figure 1B*, *Supplementary file 1*), we were able to detect *unc-17* reporter allele expression only in AWA (*Figure 4C*, *Supplementary file 2*).

Conversely, a few neurons display weak expression with previous multicopy, fosmid-based reporter constructs (RIB, AVG, PVN) (*Pereira et al., 2015*), but show no CeNGEN scRNA support for such

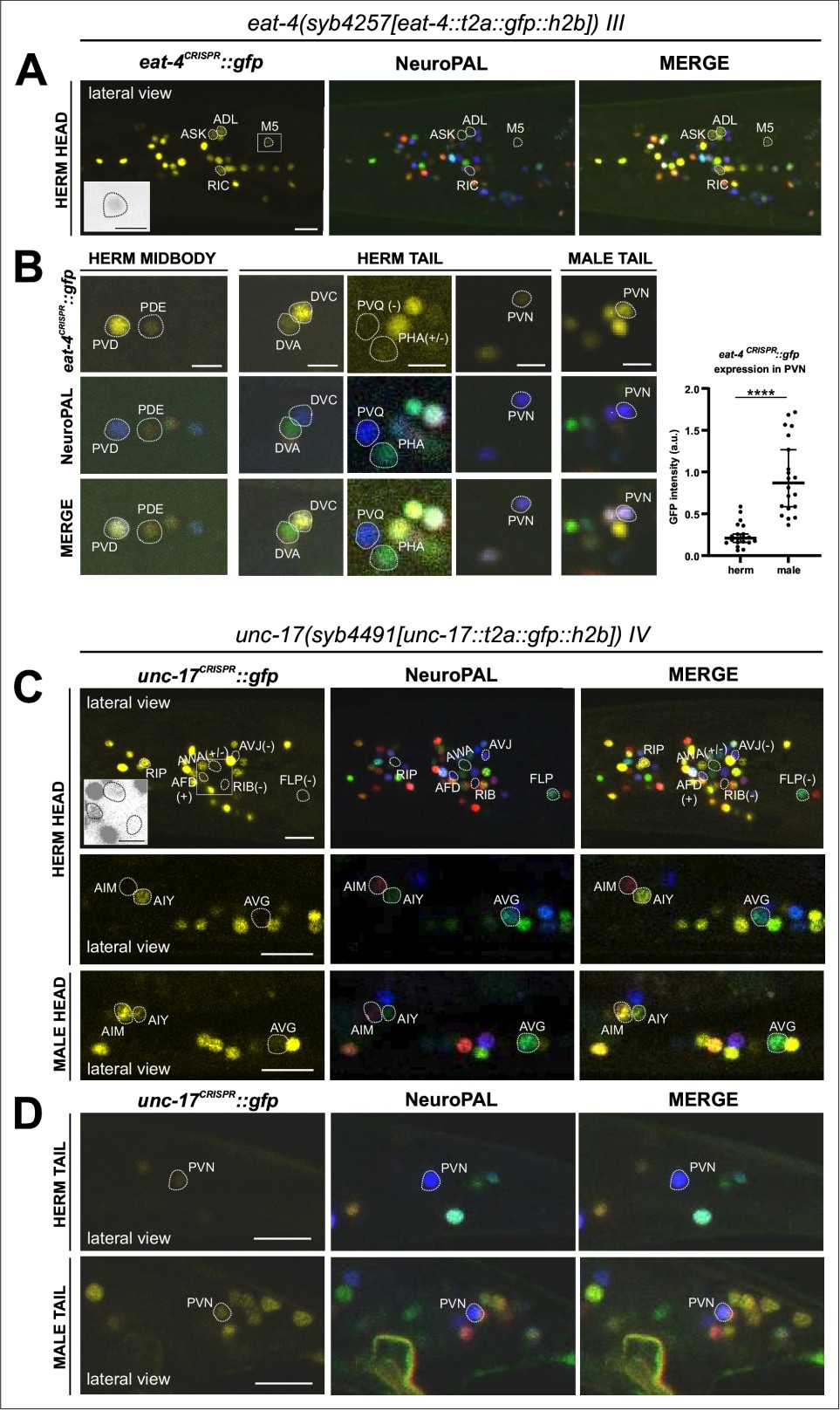

**Figure 4.** Expression of *eat-4/VGLUT* and *unc-17/VAChT* reporter alleles in the adult hermaphrodite. Neuronal expression of *eat-4(syb4257)* and *unc-17(syb4491)* was characterized with landmark strain NeuroPAL (*otIs696* and *otIs669,* respectively). Only selected neurons are shown for illustrating updates from previous reports. See **Supplementary file 2** for a complete list of neurons. (**A**) Dim expression of *eat-4(syb4257)* in head neurons

*Figure 4 continued on next page*

*Figure 4 continued*

ASK and ADL is consistent with previous fosmid-based reporter expression. RIC expression is consistent with previous observation using the same reporter allele (*Reilly et al., 2022*). In addition, dim expression is detected in pharyngeal neuron M5 (also in grayscale inset), previously not detected with *eat-4* GFP fosmid-based reporter (*otIs388*) but visible with *eat-4* mCherry fosmid-based reporter (*otIs518*). (**B**) Previously uncharacterized *eat-4* expression in PDE and DVA neurons is detected with the *eat-4(syb4257)* reporter allele. Variable expression in PHA is also occasionally detected. No expression is detected in PVQ. Expression in PVN is detected in both sexes but at a much higher level in the male. (**C**) In the head, prominent expression of *unc-17(syb4491)* in RIP and dim expression in AWA and AFD neurons are detected. There is no visible expression in RIB, FLP, or AVJ. Consistent with previous reports, AIM expresses *unc-17* only in males and not hermaphrodites. In addition, very dim expression of AVG can be detected occasionally in hermaphrodites (representative image showing an animal with no visible expression) and slightly stronger in males (representative image showing an animal with visible expression). Inset, grayscale image showing dim expression for AWA and AFD and no expression for RIB. (**D**) In the tail, PVN expresses *unc-17(syb4491)* in both sexes, consistent with previous reports. Scale bars, 10 μm in color images in A, C, and D; 5 μm in B and all grayscale images. Quantification in B is done by normalizing fluorescent intensity of *eat-4* GFP to that of the blue channel in the NeuroPAL background. Statistics, Mann-Whitney test.

expression (*Taylor et al., 2021*). The *unc-17(syb4491)* reporter allele confirmed weak but consistent expression in the PVN neurons as well as variable, borderline expression in AVG (*Figure 4C and D*). However, we failed to detect *unc-17(syb4491)* reporter allele expression in the RIB neurons.

We detected another novel site of *unc-17* expression, albeit dim, in the glutamatergic AFD neurons (*Figure 4C*, *Supplementary file 2*). This expression was not reported with previous fosmid-based reporter or CeNGEN scRNA data. Consistent with AFD and PVN being potentially cholinergic, scRNA transcript reads for *cha-1/ChAT*, the ACh-synthesizing choline acetyltransferase, were also detected in AFD and PVN (*Supplementary file 1*).

Lastly, another notable observation is the lack of any *unc-17* reporter expression or CeNGEN scRNA transcripts in the interneuron AVJ, but presence of CeNGEN scRNA transcript reads for *cha-1/ChAT* (*Supplementary file 1*), which shares exons with the *unc-17/VAChT* locus (*Alfonso et al., 1994*). Although no reporter data is available for *cha-1/ChAT*, such interesting mismatch between available *unc-17* and *cha-1/ChAT* expression data could provide a hint to potential non-vesicular cholinergic transmission in the AVJ neurons in *C. elegans*, potentially akin to reportedly non-vesicular release of acetylcholine in the visual system of *Drosophila* (*Yang and Kunes, 2004*).

## Expression of reporter alleles for GABAergic pathway genes in the hermaphrodite

### Expression of *unc-25/GAD*

The most recent analysis of GABAergic neurons identified GABA-synthesizing cells by anti-GABA staining and an SL2-based *unc-25/GAD* reporter allele that monitors expression of the rate-limiting step of GABA synthesis, generated by CRISPR/Cas9 engineering (*Gendrel et al., 2016*). The CeNGEN scRNA atlas shows robust support for these assignments at all four threshold levels (*Figure 1B*, *Supplementary file 1*). *unc-25* scRNA signals (but no reporter signals) were detected at several orders of magnitude lower levels in three additional neuron classes (AWA, AVH, PVT), but only with the least robust threshold level.

In this study we generated another *unc-25/GAD* reporter allele, using a *t2a::gfp::h2b* cassette (*ot1372*) (*Figure 2*). This allele showed the same expression pattern as the previously described SL2-based *unc-25(ot867)* reporter allele (*Figure 5A*, *Supplementary file 2*). This includes a lack of expression in a number of neurons that stain with anti-GABA antibodies (SMD, AVA, AVB, AVJ, ALA, and AVF) and GLR glia, corroborating the notion that these neurons and glia take up GABA from other cells (indeed, a subset of those cells do express the GABA uptake reporter SNF-11; *Gendrel et al., 2016*).

We carefully examined potential *unc-25/GAD* reporter allele expression in the AMsh glia, which were reported to generate GABA through *unc-25/GAD* (*Duan et al., 2020*; *Fernandez-Abascal et al., 2022*). We did not detect visible *unc-25(ot867)* or *unc-25(ot1372)* reporter allele expression in AMsh, consistent with the failure to directly detect GABA in AMsh through highly sensitive anti-GABA staining (*Gendrel et al., 2016*). Since these reporters do not capture an alternatively spliced isoform b.1 (https://www.wormbase.org), we generated another reporter allele, *unc-25(ot1536)*, to specifically

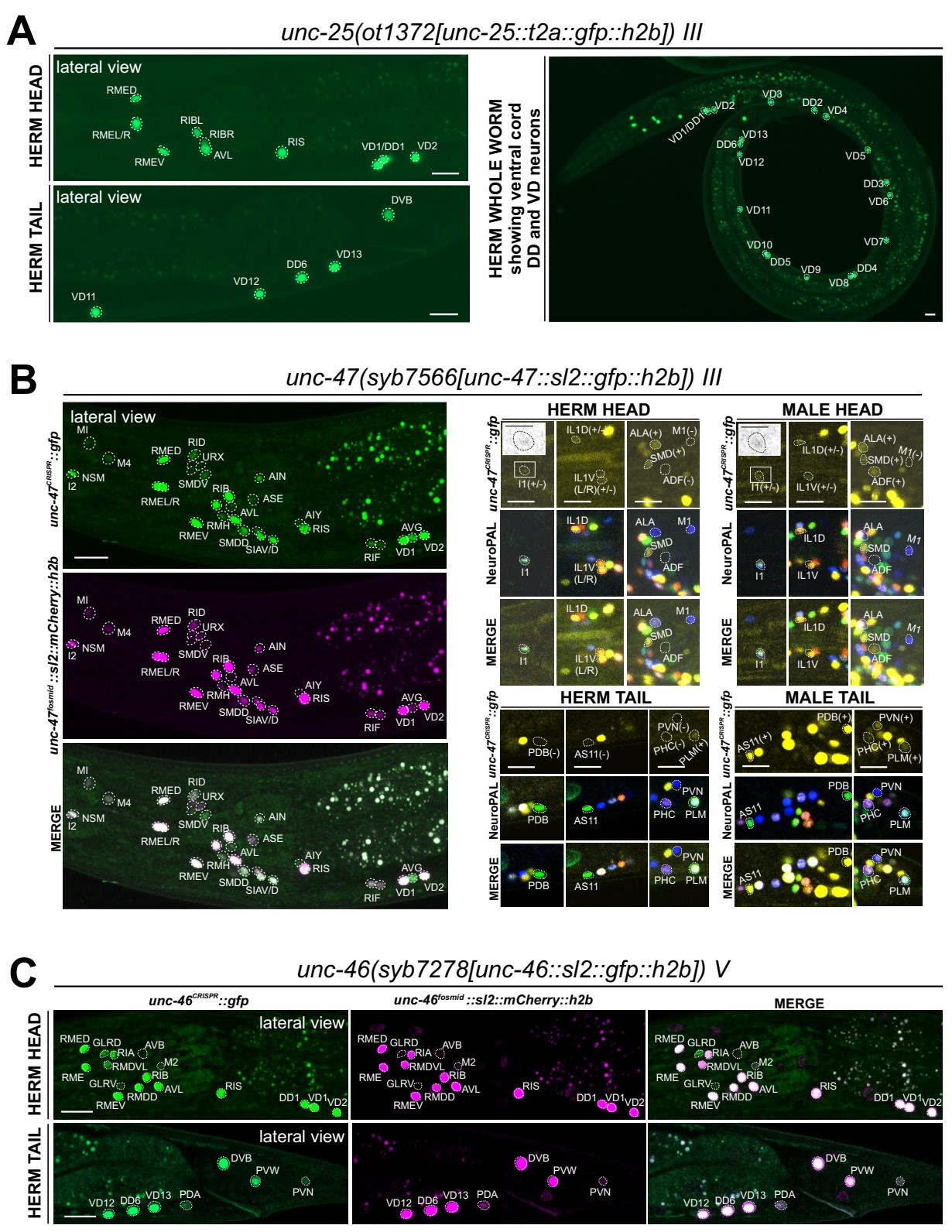

**Figure 5.** Expression of GABA pathway genes in the adult hermaphrodite. (**A**) Expression of the *unc-25/GAD* reporter allele *ot1372* is detected in the head, ventral nerve cord, and tail neurons. The expression pattern of this new T2A-based reporter allele is similar to that of a previously described SL2-based reporter allele, *unc-25(ot867)* (**Gendrel et al., 2016**). (**B**) Expression of *unc-47/VGAT* reporter allele *syb7566*. Left, the expression pattern of the reporter allele largely matches that of a previously described *unc-47* mCherry fosmid-based reporter (*otIs564*) in the head. Right, a close-up view for the

*Figure 5 continued on next page*

*Figure 5 continued*

characterization of the reporter allele expression with landmark strain NeuroPAL (*otIs669*). In the head, consistent with previous reports of the *unc-47* fosmid-based reporter (*otIs564*), dim expression of *unc-47(syb7566)* in SMD, ALA, and very dim and variable expression in IL1 is detected in both sexes, and *unc-47(syb7566)* is expressed in ADF only in the male and not hermaphrodite. In addition, the reporter allele is also expressed at a very dim level in the pharyngeal neuron I1 (also in inset) whereas no expression is detected in M1. In the tail, consistent with previous reports of the fosmid, sexually dimorphic expression of the *unc-47(syb7566)* reporter allele is also detected in PDB, AS11, PVN, and PHC only in the male and not the hermaphrodite. In addition, we also detected very dim expression of PLM in both sexes, confirming potential dim expression of the *unc-47* mCherry fosmid-based reporter that was only readily visible after anti-mCherry staining in the past (*Serrano-Saiz et al., 2017b*). Scale bars, 5 µm for insets and 10 µm for all other images. (**C**) Expression of *unc-46/LAMP* reporter allele *syb7278* is largely similar to that of the previously described *unc-46/LAMP* mCherry fosmid-based reporter (*otIs568*). We also observed expression of both the reporter allele and fosmid-based reporter in PVW, PVN, and very dimly in PDA. Scale bars, 10 µm.

target this isoform. However, we did not observe any discernible fluorescent reporter expression from this allele. Hence, it is unlikely that an alternative isoform could contribute to expression in additional cell types.

## Expression of *unc-47/VGAT*

While promoter-based transgenes for the vesicular transporter for GABA, *unc-47/VGAT*, had shown expression patterns that precisely match that of *unc-25/GAD* (*Eastman et al., 1999*), we had noted in our previous analysis of the GABA system that a fosmid-based reporter showed much broader expression in many additional neuron classes that showed no sign of GABA usage (*Gendrel et al., 2016*). In several of these neuron classes both the fosmid-based reporter and the CeNGEN scRNA data indicate very robust expression (e.g. AIN, SIA, SDQ), while in many others scRNA transcripts are only evident at looser thresholds and, correspondingly, fosmid-based reporter expression in these cells is often weak (*Supplementary file 1*; *Gendrel et al., 2016*). To investigate this matter further, we CRISPR/Cas9-engineered a *gfp*-based reporter allele for *unc-47*, *syb7566*, and first crossed it with an mCherry-based *unc-47* fosmid-based reporter (*otIs564*) as a first-pass assessment for any obvious overlaps and mismatches of expression patterns between the two (*Figure 5B*, left side panels). The vast majority of neurons exhibited overlapping expression between *syb7566* and *otIs564*. There were also many notable similarities in the robustness of expression of the fosmid-based reporter and the reporter allele (*Supplementary file 1*). In a few cases where the fosmid-based reporter expression was so dim that it is only detectable via antibody staining against its fluorophore (mCherry) (*Gendrel et al., 2016*; *Serrano-Saiz et al., 2017b*), the reporter allele expression was readily visible (*Supplementary file 1*).

The very few mismatches of expression of the fosmid-based reporter and the reporter allele included the pharyngeal neuron M1, which expresses no visible *unc-47(syb7566)* reporter allele but weak fosmid-based reporter expression, and the pharyngeal neuron I1, which expresses dim *syb7566* but no fosmid-based reporter (*Figure 5B*, right side panels). AVJ shows very dim and variable *unc-47(syb7566)* reporter allele expression but no fosmid-based reporter expression. Since AVJ stains with anti-GABA antibodies (*Gendrel et al., 2016*), this neuron likely engages in vesicular relase of GABA, even though its source of GABA remains unclear since it neither expresses conventional GABA synthesis machinery (UNC-25/GAD) nor GABA uptake machinery (SNF-11). Other neurons previously shown to stain with anti-GABA antibodies and to express the *unc-47* fosmid-based reporter (ALA and SMD) (*Gendrel et al., 2016*) still show expression of the *unc-47* reporter allele.

In addition, while the reporter allele of *unc-47/VGAT*, in conjunction with CeNGEN scRNA data, corroborates the notion that *unc-47/VGAT* is expressed in all GABA-synthesizing and most GABA uptake neurons, there is a substantial number of *unc-47*-positive neurons that do not show any evidence of GABA presence. This suggests that UNC-47/VGAT may transport another unidentified neurotransmitter (see Discussion) (*Gendrel et al., 2016*).

## Expression of *unc-46/LAMP*

In all GABA-synthesizing neurons, the UNC-47/VGAT protein requires the LAMP-like protein UNC-46 for proper localization (*Schuske et al., 2007*). A previously analyzed fosmid-based reporter confirmed *unc-46/LAMP* expression in all 'classic' GABAergic neurons (i.e. anti-GABA and *unc-25/GAD*-positive neurons), but also showed robust expression in GABA- and *unc-47*-negative neurons, such as RMD (*Gendrel et al., 2016*). This non-GABAergic neuron expression is confirmed by CeNGEN scRNA

data (*Taylor et al., 2021*; *Supplementary file 1*). We generated an *unc-46/LAMP* reporter allele, *syb7278*, and found its expression to be largely similar to that of the fosmid-based reporter and to the scRNA data (*Figure 5C*, *Supplementary file 1*), therefore corroborating the non-GABAergic neuron expression of *unc-46/LAMP*. We also detected previously unreported expression in the PVW and PVN neurons in both the reporter allele and fosmid-based reporter (*Figure 5C*), thereby further corroborating CeNGEN data. In addition, we also detected very dim expression in PDA (*Figure 5C*), which shows no scRNA transcript reads (*Supplementary file 1*). With one exception (pharyngeal M2 neuron class), the sites of non-GABAergic neuron expression of *unc-46/LAMP* expression do not show any overlap with the sites of *unc-47/VGAT* expression, indicating that these two proteins have functions independent of each other.

## Expression of reporter alleles for serotonin biosynthetic enzymes, *tph-1/TPH* and *bas-1/AAAD*, in the hermaphrodite

*tph-1/TPH* and *bas-1/AAAD* code for enzymes required for serotonin (5-HT = 5-hydroxytryptamine) synthesis (*Figure 1A*). scRNA transcripts for *tph-1* and *bas-1* are detected in previously defined serotonergic neurons at all four threshold levels (HSN, NSM, ADF) (*Figure 1*, *Supplementary file 1*). In addition to these well-characterized sites of expression, several of the individual genes show scRNA-based transcripts in a few additional cells: *tph-1* at all four threshold levels in AFD and MI. Neither of these cells display scRNA transcripts for *bas-1/AAAD*, the enzyme that metabolizes the TPH-1 product 5-HTP (5-hydroxytryptophan) into serotonin (5-HT) (*Figure 1A*). To further investigate these observations, we generated reporter alleles for both *tph-1* and *bas-1* (*Figure 2*). Expression of the *tph-1* reporter allele *syb6451* confirmed expression in the previously well-described neurons that stained positive for serotonin, namely NSM, HSN, and ADF, matching CeNGEN data. While expression in AFD (seen at all four threshold levels in the CeNGEN scRNA atlas) could not be confirmed with the reporter allele, expression in the pharyngeal MI neurons could be confirmed (*Figure 6A*, *Figure 6—figure supplement 1*, *Supplementary file 2*).

We detected co-expression of the *bas-1* reporter allele, *syb5923*, with *tph-1(syb6451)* in NSM, HSN, and ADF, in accordance with the previous reporter and scRNA data (*Figure 6B*, *Supplementary file 2*). However, *bas-1(syb5923)* is not co-expressed with *tph-1* in MI (*Figure 6A and B*), nor is there CeNGEN-transcript evidence for *bas-1/AAAD* in MI (*Figure 1*, *Supplementary file 1*). Hence, TPH-1-synthesized 5-HTP in MI is not metabolized into 5-HT (serotonin), consistent with the lack of serotonin-antibody staining in MI (*Horvitz et al., 1982*; *Sze et al., 2000*).

We also detected *tph-1(syb6451)* reporter allele expression in the serotonergic VC4 and VC5 neurons (*Figure 6A*, *Supplementary file 2*), consistent with scRNA data (*Figure 1*, *Supplementary file 1*) and previous reporter transgene data (*Mondal et al., 2018*). This suggests that these neurons are capable of producing 5-HTP. However, there is no *bas-1(syb5923)* expression in VC4 or VC5, consistent with previous data showing that serotonin is taken up, but not synthesized by them (*Duerr et al., 2001*) (more below on monoamine uptake; *Tables 1 and 2*).

As expected from the role of *bas-1/AAAD* in dopamine synthesis (*Hare and Loer, 2004*), *bas-1(syb5923)* is also expressed in dopaminergic neurons PDE, CEP, and ADE. In addition, it is also expressed weakly in URB, consistent with scRNA data. We did not detect visible expression in PVW or PVT, both of which showed very low levels of scRNA transcripts (*Figure 1*, *Supplementary file 1*). Expression of *bas-1/AAAD* in URB may suggest that URB generates a non-canonical monoamine (e.g. tryptamine, phenylethylamine [PEA], or histamine), but since URB expresses no vesicular transporter (*cat-1/VMAT*, see below), we consider it unlikely that any such monoamine would be secreted via canonical vesicular synaptic release mechanisms.

## Expression of a reporter allele of *cat-2/TH*, a dopaminergic marker, in the hermaphrodite

The CeNGEN scRNA atlas shows transcripts for the rate-limiting enzyme of dopamine synthesis encoded by *cat-2/TH* (*Figure 1B*, *Supplementary file 1*) at all four threshold levels in all three previously described dopaminergic neuron classes in the hermaphrodite, ADE, PDE, and CEP (*Sulston et al., 1975*; *Sulston et al., 1980*; *Lints and Emmons, 1999*). At lower threshold levels, transcripts can also be detected in the OLL neurons. A CRISPR/Cas9-engineered reporter allele for *cat-2/TH*, *syb8255*, confirmed expression in ADE, PDE, and CEP in adult hermaphrodites (*Figure 7A*, *Supplementary file*

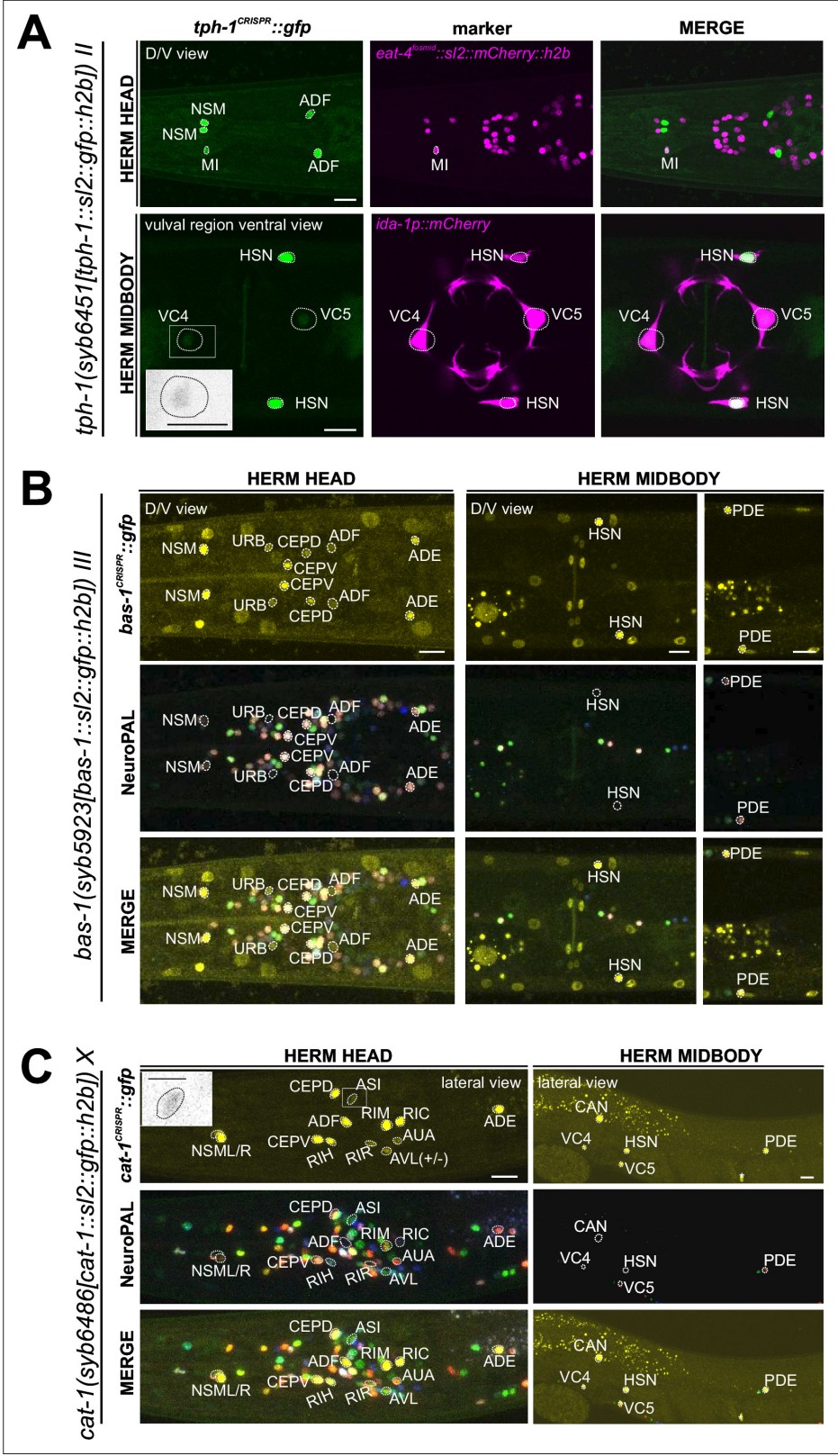

**Figure 6.** Expression of *tph-1/TPH*, *bas-1/AAAD*, and *cat-1/VMAT* reporter alleles in the adult hermaphrodite. (**A**) Dorsoventral view of a hermaphrodite head and midbody expressing *tph-1(syb6451)*. *tph-1* expression is detected robustly in the MI neuron and dimly and variably in VC4 and VC5. Neuron identities for MI and VC4 and VC5 were validated using *otIs518[eat-4(fosmid)::sl2::mCherry::h2b]* and *vsIs269[ida-1::mCherry]*, respectively, as

*Figure 6 continued on next page*

*Figure 6 continued*

landmarks. Inset, grayscale image highlighting dim expression in VC4. Larval expression of this reporter allele is shown in *Figure 6—figure supplement 1*. (**B**) Neuronal expression of *bas-1(syb5923)* characterized with the landmark NeuroPAL (*otIs669*) strain in the head and midbody regions of young adult hermaphrodites. Dorsoventral view of the adult head shows *bas-1/AAAD* expression in left-right neuron pairs, including previously reported expression in NSM, CEP, ADF, and ADE (*Hare and Loer, 2004*). Additionally, we observed previously unreported expression in the URB neurons. Non-neuronal *bas-1/AAAD* expression is detected in other non-neuronal cell types as reported previously (*Yu et al., 2023*; also see *Figure 14—figure supplement 1*, *Figure 14*). (**C**) Lateral views of young adult hermaphrodite head and midbody expressing *cat-1/VMAT* (*syb6486*). Previously unreported *cat-1/VMAT* expression is seen in RIR, CAN, AUA, ASI (also in inset), and variably, AVL. Non-neuronal expression of *cat-1/VMAT* is detected in a single midbody cell in the gonad (also see *Figure 14—figure supplement 1*), marked with an asterisk. Scale bars, 10 μm for all color images; 5 μm for the inset in grayscale.

The online version of this article includes the following figure supplement(s) for figure 6:

**Figure supplement 1.** *tph-1/TPH* reporter allele expression in the hermaphrodite larvae.

---

*2*). As expected and described above, all three neuron classes also expressed *bas-1/AAAD* (*Figure 6B*) and *cat-1/VMAT* (*Figure 6C*, see below) (*Supplementary file 2*). We did not detect visible expression of *cat-2(syb8255)* in OLL. The OLL neurons also display no scRNA transcripts or reporter allele expression of *bas-1/AAAD* or *cat-1/VMAT*. No additional sites of expression of *cat-2(syb8255)* were detected in the adult hermaphrodite.

## Expression of reporter alleles of *tdc-1/TDC* and *tbh-1/TBH*, markers for tyraminergic and octopaminergic neurons, in the hermaphrodite

The invertebrate analogs of adrenaline and noradrenaline, tyramine and octopamine, are generated by *tdc-1* and *tbh-1* (*Figure 1A*; *Alkema et al., 2005*). Previous work had identified expression of *tdc-1* in the hermaphrodite RIM and RIC neurons and *tbh-1* in the RIC neurons (*Alkema et al., 2005*). Transcripts in the CeNGEN atlas match those sites of expression for both *tdc-1* (scRNA at four threshold levels in RIM and RIC neurons) and *tbh-1* (scRNA at four threshold levels in RIC neurons) (*Figure 1B*, *Supplementary file 1*). Much lower transcript levels are present in a few additional, non-overlapping neurons (*Figure 1B*). CRISPR/Cas9-engineered reporter alleles confirmed *tdc-1* expression in RIM and RIC and *tbh-1* expression in RIC (*Figure 7B and C*, *Supplementary file 2*). In addition, we also detected dim expression of *tbh-1(syb7786)* in all six IL2 neurons, corroborating scRNA transcript data (*Figure 7C*, *Supplementary files 1 and 2*). However, IL2 neurons do not exhibit expression of the reporter allele of *tdc-1*, which acts upstream of *tbh-1* in the octopamine synthesis pathway, or of *cat-1/VMAT*, the vesicular transporter for octopamine (*Figure 6C*, see below). Hence, the IL2 neurons are unlikely to produce or synaptically release octopamine, but they may produce another monoaminergic signal (*Table 2*).

## Expression of a reporter allele of *cat-1/VMAT*, a marker for monoaminergic identity, in the hermaphrodite

As the vesicular monoamine transporter, *cat-1/VMAT* is expected to be expressed in all neurons that synthesize serotonin, dopamine, tyramine, and octopamine (*Figure 1A*). Both scRNA data and a CRISPR/Cas9-engineered reporter allele, *syb6486*, confirm expression in all these cells (*Figure 6C*, *Supplementary file 2*). In addition, based on antibody staining and previous fosmid-based reporters, *cat-1/VMAT* is known to be expressed in neurons that do not synthesize serotonin but are nevertheless positive for serotonin antibody staining (VC4, VC5, and RIH) (*Duerr et al., 1999*; *Duerr et al., 2001*; *Serrano-Saiz et al., 2017b*). Again, both scRNA data and a CRISPR/Cas9-engineered reporter allele, *syb6486*, confirm expression in these cells (*Figure 6C*, *Supplementary file 2*).

In addition to these canonical monoaminergic neurons, the CeNGEN scRNA data shows *cat-1/VMAT* expression at all four threshold levels in RIR, CAN, AVM and, at a much lower threshold, eight additional neuron classes (*Figure 1B*, *Supplementary file 1*). Our *cat-1/VMAT* reporter allele, *syb6486*, corroborates expression in RIR and CAN, but not in AVM (*Figure 6C*, *Supplementary file 2*). We also observed expression of the *cat-1* reporter allele in two of the neuron classes with scRNA transcripts at the lowest threshold level, ASI and variably, AVL (*Figure 6C*, *Supplementary file 1*). Interestingly, AVL does not express any other monoaminergic pathway genes (*Supplementary file 2*), therefore it

**Table 1.** Neurons that uptake monoaminergic neurotransmitters.

+: presence of reporter allele expression; -: lack of visible reporter allele expression; +/-: dim and variable expression (variability is only detected when reporter fluorescent intensity is low); m: anti-serotonin staining observed in males; *: sex-specific neurons; **: variable/very dim antibody staining reported in previous publications. ***N/A=not presently applicable because betaine is provided by diet, in addition to possible endogenous synthesis. See text for citations.

|  | | Uptake | Synthesis | Release |
|---|---|---|---|---|
|  | Neuron | *mod-5* | *tph-1* | *cat-1* |
|  | ADF | + | + | + |
|  | AIM | + | - | - |
|  | I5** | - | - | +/- |
|  | NSM | + | + | + |
|  | PVW(m)** | - | - | - |
|  | RIH | + | - | + |
|  | URX** | +/- | - | - |
|  | *HSN | - | + | + |
|  | *VC4-5** | - | +/- | + |
|  | *CEM** | + | + | - |
|  | *CP1-6 | + | + | + |
|  | *PGA | + | - | + |
|  | *R1B | - | + | + |
|  | *R3B | + | + | + |
| Serotonin | *R9B | + | + | + |
|  | Neuron | *oct-1* | *tdc-1* | *cat-1* |
| Tyramine | RIM | + | + | + |

*Table 1 continued on next page*

*Table 1 continued*

| Neuron | Uptake *snf-3* | Synthesis N/A*** | Release *cat-1* |
|---|---|---|---|
| AUA | + | | + |
| CAN | + | | + |
| NSM | +/- | | + |
| RIM | + | | + |
| RIR | +/- | | + |
| ASI | +/- | | + |
| M3 | +/- | | - |
| AIB | + | | - |
| DVB | +/- | | - |
| SMD | +/- | | - |
| RIS | + | | - |
| URX | +/- | | - |
| PDA | +/- | | - |
| ASG | +/- | | - |
| DA9 | +/- | | - |
| VB1-11 | +/- | | - |
| PHC | + | | - |
| PVN | + | | - |
| VA12 | +/- | | - |
| RMH | +/- | | - |
| *PDC | + | | + |
| *PHD | + | | - |
| Betaine | *PVV +/- | | - |

may be transporting a new amine yet to be discovered. This scenario also applies for two male-specific neurons (more below). As previously mentioned, we detected no *cat-1/VMAT* expression in the *tph-1/TPH*-positive MI or the *cat-2/TH*-positive OLL neurons.

The *cat-1/VMAT* reporter allele revealed expression in an additional neuron class, the AUA neuron pair (**Figure 6C**, **Supplementary file 2**). Expression in this neuron is not detected in scRNA data; however, such expression may be consistent with previous CAT-1/VMAT antibody staining data (**Duerr et al., 1999**). These authors found the same expression pattern as we detected with *cat-1/VMAT* reporter allele, except for the AIM neuron, which Duerr et al. identified as CAT-1/VMAT antibody-staining positive. However, neither our reporter allele, nor a fosmid-based *cat-1/VMAT* reporter, nor scRNA data showed expression in AIM, and we therefore think that the neurons identified by Duerr et al. as AIM may have been the AUA neurons instead (see also **Serrano-Saiz et al., 2017b**). Additionally, a *cat-1*-positive neuron pair in the ventral ganglion, unidentified but mentioned by **Duerr et al., 1999**, is likely the tyraminergic RIM neuron pair, based on our reporter allele and CeNGEN scRNA data.

## Expression of reporter alleles of monoamine uptake transporters in the hermaphrodite

In addition to or in lieu of synthesizing monoamines, neurons can uptake them from their surroundings. To investigate the cellular sites of monoamine uptake in more detail, we analyzed fluorescent protein expression from engineered reporter alleles for the uptake transporter of serotonin (*mod-5/*

**Table 2.** Categories of neuronal expression patterns for monoaminergic neurotransmitter pathway genes.
Criteria for monoaminergic neurotransmitter assignment and a summary for neurons with updated identities are presented here. The categories represent our best assessments based on available data; in every category there is a possibility for the existence of non-canonical synthesis and/or uptake mechanisms that are yet to be discovered. +: presence of reporter allele expression (incl. dim); -: lack of visible reporter allele expression; *bas-1*-dependent unknown monoamine?=*bas-1*-dependent unknown monoamine (histamine, tryptamine, PEA; see *Figure 1—figure supplement 1A* and Discussion); unknown monoamine?=potentially non-canonical monoamines; see Discussion and Results sections on specific gene expression patterns; 5-HT=5-hydroxytryptamine, or serotonin; 5-HTP=5-hydroxytryptophan; PEOH = β-hydroxyphenethylamine, or phenylethanolamine; *: The expression of *tph-1* in VC4-5, *bas-1* in R4B and R6B, *cat-1* in AVL, and *snf-3* in NSM, RIR, ASI, URX, M3, DVB, SMD, PDA, ASG, DA9, VA12, VB1-11, RMH, and PVV is dim and variable (this study; variability is only detected when reporter fluorescent intensity is low); anti-5-HT staining in VC4, VC5, CEM, I5, URX, and PVV (male) is variable in previous reports (see text for citations). ** indicates that R4B and R7B express 5-HT synthesis machinery (*tph-1* and *bas-1*), but do not stain with 5-HT antibodies.

| Synthesis (and/or uptake) | cat-1 | tph-1 | cat-2 | bas-1 | tdc-1 | tbh-1 | mod-5 | snf-3 | oct-1 | Direct staining | Sex-specific neurons | Sex-shared neurons |
|---|---|---|---|---|---|---|---|---|---|---|---|---|
| Tyramine+*bas-1*-dependent unknown monoamine? | + | - | - | + | + | - | - | - | - | | HOA | |
| Tyramine+*bas-1*-dependent unknown monoamine? | - | - | - | + | + | - | - | - | - | | R8A | |
| Tyramine+dopamine | + | - | + | + | + | - | - | - | - | Dopamine | R7A | |
| Tyramine (+uptake)+betaine (uptake) | + | - | - | - | + | - | - | + | + | | | RIM |
| *bas-1*-dependent unknown monoamine? | + | - | - | + | - | - | - | - | - | | R2A | |
| *bas-1*-dependent unknown monoamine? | - | - | - | + | - | - | - | - | - | | R3A, R6A, R6B*, PCB, SPC, DVE, DVF | URB |
| Octopamine | + | - | - | - | + | + | - | - | - | | | RIC |
| Octopamine | - | - | - | - | + | + | - | - | - | | R8B | |
| Dopamine | + | - | + | + | - | - | - | - | - | Dopamine | R5A, R9A | ADE, CEP, PDE |
| 5-HTP (synthesis)+5-HT (alternative synthesis/uptake mechanism?)+unknown monoamine? | - | + | - | - | - | + | - | - | - | 5-HT | CEM* | |
| 5-HTP | - | + | - | - | - | - | - | - | - | | | MI |
| PEOH? | - | - | - | + | - | + | - | - | - | | R2B | |
| 5-HT+PEOH? | - | + | - | + | - | + | - | - | - | | R7B** | |
| 5-HT+PEOH? | + | + | - | + | - | + | - | - | - | 5-HT | R1B | |
| 5-HT+PEOH? | + | + | - | + | - | + | + | - | - | 5-HT | R3B | |
| 5-HT+PEOH? | + | + | - | - | - | + | - | - | - | | R4B** | |
| 5-HT (uptake) | + | - | - | - | - | - | + | - | - | 5-HT | PGA | RIH |
| 5-HT (uptake) | - | - | - | - | - | - | + | - | - | 5-HT | | AIM |
| 5-HT (uptake)+betaine (uptake) | - | - | - | - | - | - | + | + | - | 5-HT | | URX* |
| 5-HT (& uptake) | + | + | - | + | - | - | + | - | - | 5-HT | CP1-6 | ADF |
| 5-HT (alternative synthesis/uptake mechanism?) | - | - | - | - | - | - | - | - | - | 5-HT | | I5*, PVV (male only) |
| 5-HT | + | + | - | + | - | - | - | - | - | 5-HT | HSN | |
| 5-HTP (synthesis) and 5-HT (uptake) | + | + | - | - | - | + | + | - | - | 5-HT | R9B | |
| 5-HTP (synthesis) and 5-HT (alternative synthesis/uptake mechanism?) | + | + | - | - | - | - | - | - | - | 5-HT | VC4-5* | |

*Table 2 continued on next page*

Table 2 continued

| Synthesis (and/or uptake) | cat-1 | tph-1 | cat-2 | bas-1 | tdc-1 | tbh-1 | mod-5 | snf-3 | oct-1 | Direct staining | Sex-specific neurons | Sex-shared neurons |
|---|---|---|---|---|---|---|---|---|---|---|---|---|
| Unknown monoamine? | + | - | - | - | - | - | - | - | - | | PVX, PVY | AVL* |
| Unknown monoamine? | - | - | - | - | - | + | - | - | - | | HOB, R5B | IL2 |
| 5-HT+betaine (uptake) | + | + | - | + | - | - | - | + | - | 5-HT | | NSM* |
| Betaine (uptake) | + | - | - | - | - | - | - | + | - | | PDC | AUA, CAN, RIR*, ASI* |
| Betaine (uptake) | - | - | - | - | - | - | - | + | - | | PHD, PVV* | M3*, AIB, DVB*, SMD*, RIS, PDA*, ASG*, DA9*, PHC, PVN, VA12*, VB1-11*, RMH* |

SERT(vlc47)), the predicted uptake transporter for tyramine (oct-1/OCT(syb8870)), and that for betaine (snf-3/BGT1(syb7290)).

## Serotonin/5-HT uptake

Using a promoter-based transgene and antibody staining, previous work had shown expression of the serotonin uptake transporter mod-5/SERT in NSM, ADF, RIH, and AIM (Jafari et al., 2011; Maicas et al., 2021). This matched the observations that RIH and AIM do not synthesize serotonin (i.e. do not express tph-1), but stain positive with a serotonin antibody (Jafari et al., 2011). In mod-5 mutants or wild type worms treated with serotonin reuptake inhibitors (such as the SSRI fluoxetine), RIH and AIM lose serotonin immunoreactivity (Jafari et al., 2011). We analyzed a CRISPR-based reporter allele, mod-5(vlc47) (Maicas et al., 2021), and confirmed expression in the four neuron classes NSM, ADF, RIH, and AIM (Figure 8). Because only NSM, ADF, and RIH, but not AIM, express the reporter allele of the monoamine transporter CAT-1/VMAT (Figure 6), AIM likely functions as a serotonin uptake/clearance neuron (Tables 1 and 2; see also Discussion). In addition, we also detected dim mod-5/SERT expression in the phasmid neuron class PHA and very dim, variable signals in URX (Figure 8A, B, E) consistent with scRNA data (Supplementary file 1). The results for anti-serotonin-staining from previous reports are variable in a few neurons, possibly due to differences in staining methods (including URX, I5, VC4, VC5, and PVW Loer and Kenyon, 1993; Rand and Nonet, 1997; Duerr et al., 1999; Serrano-Saiz et al., 2017b). In light of its mod-5/SERT reporter expression, URX may acquire serotonin via mod-5, akin to AIM (Tables 1 and 2).

In the hermaphrodite-specific neurons HSN, VC4, and VC5, we did not observe expression of the mod-5/SERT reporter allele (Tables 1 and 2). Since VC4 and VC5 do not express the complete synthesis pathway for serotonin, we infer that the anti-serotonin staining in these neurons is a result of alternative serotonin uptake or synthesis mechanisms. A similar scenario holds for the pharyngeal neuron I5, which was previously reported to stain weakly for serotonin (Serrano-Saiz et al., 2017b).

## Tyramine uptake

Biochemical studies in vertebrates have shown that the SLC22A1/2/3 (aka OCT-1/2/3) organic cation transporters can uptake monoaminergic neurotransmitters (Nigam, 2018), with SLC22A2 being apparently selective for tyramine (Berry et al., 2016). oct-1 is the ortholog of the OCT subclass of SLC22 family members (Zhu et al., 2015), but neither its expression nor function in the nervous system had been previously reported. We tagged the endogenous oct-1 locus with an sl2::gfp::h2b cassette (syb8870) and, within the nervous system, observed exclusive expression in the RIM neuron (Figure 8H and I), indicating that RIM is likely capable of uptaking tyramine in addition to synthesizing it via tdc-1/TDC. This is consistent with RIM being the only neuron showing oct-1 scRNA transcripts at all four threshold levels in the CeNGEN atlas (Supplementary file 1).

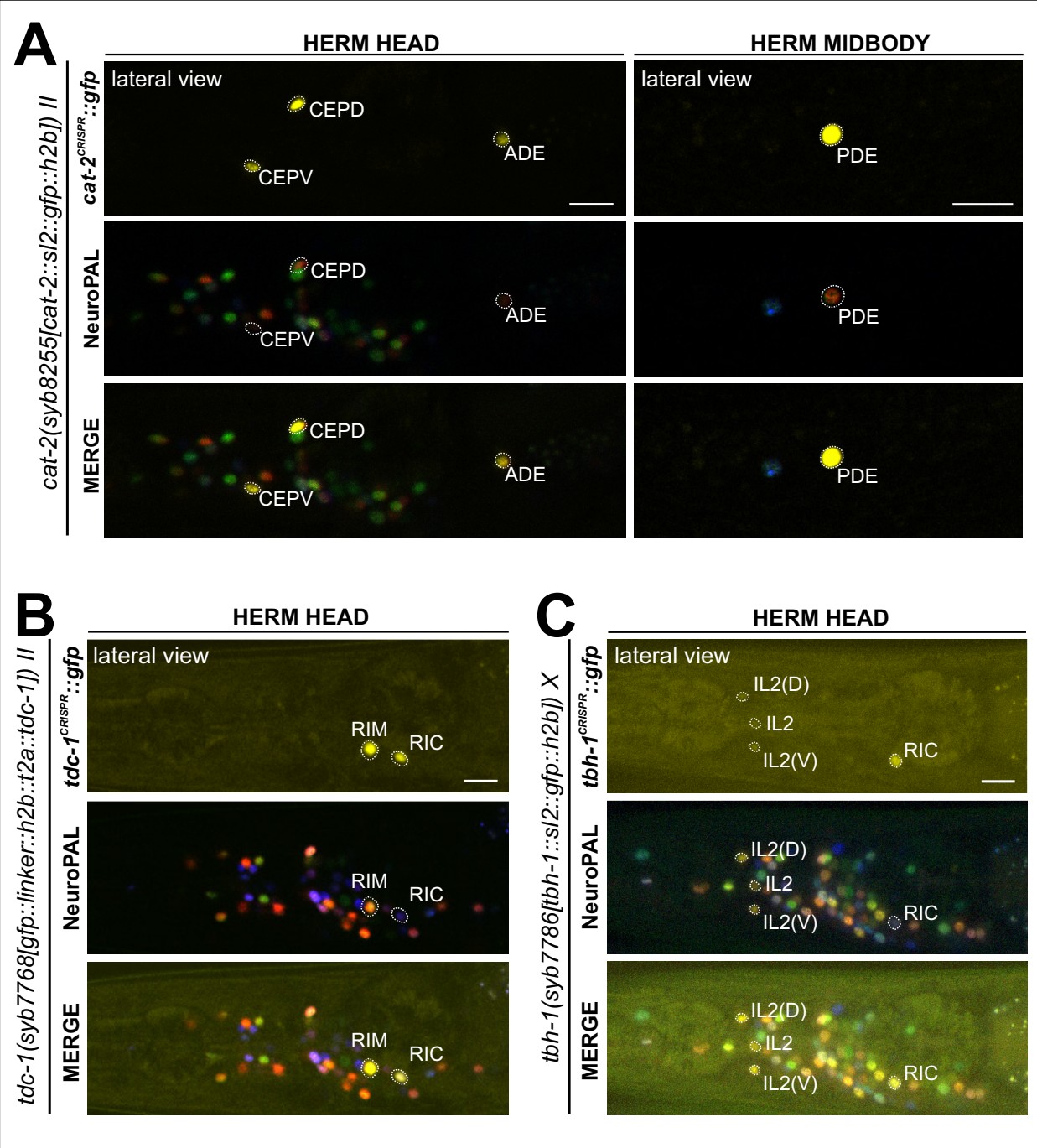

**Figure 7.** Expression of *cat-2/TH*, *tdc-1/TDC*, and *tbh-1/TBH* reporter alleles in the adult hermaphrodite. Neuronal expression was characterized with landmark strain NeuroPAL (*otIs669*). Lateral views of young adult hermaphrodites expressing reporter alleles for (**A**) *cat-2(syb8255)*, (**B**) *tbh-1(syb7786)*, and (**C**) *tdc-1(syb7768)*. (**A**) *cat-2/TH* expression in CEP, ADE, and PDE match previously reported dopamine straining expression (*Sulston et al., 1975*). (**B**) and (**C**) Head areas are shown; no neuronal expression was detected in other areas. *tdc-1* expression matches previous analysis (*Alkema et al., 2005*). We detected previously unreported expression of *tbh-1* in all six IL2 neurons at low levels. Scale bars, 10 μm.

## Betaine uptake

Notably, four CAT-1/VMAT-expressing neuron classes, CAN, AUA, RIR, and ASI, do not express biosynthetic enzymes for synthesis or uptake transporters of the four conventional monoaminergic transmitters known to be employed in *C. elegans* (serotonin, dopamine, octopamine, or tyramine). Hence, these neuron classes might instead synthesize or uptake another transmitter for ensuing synaptic

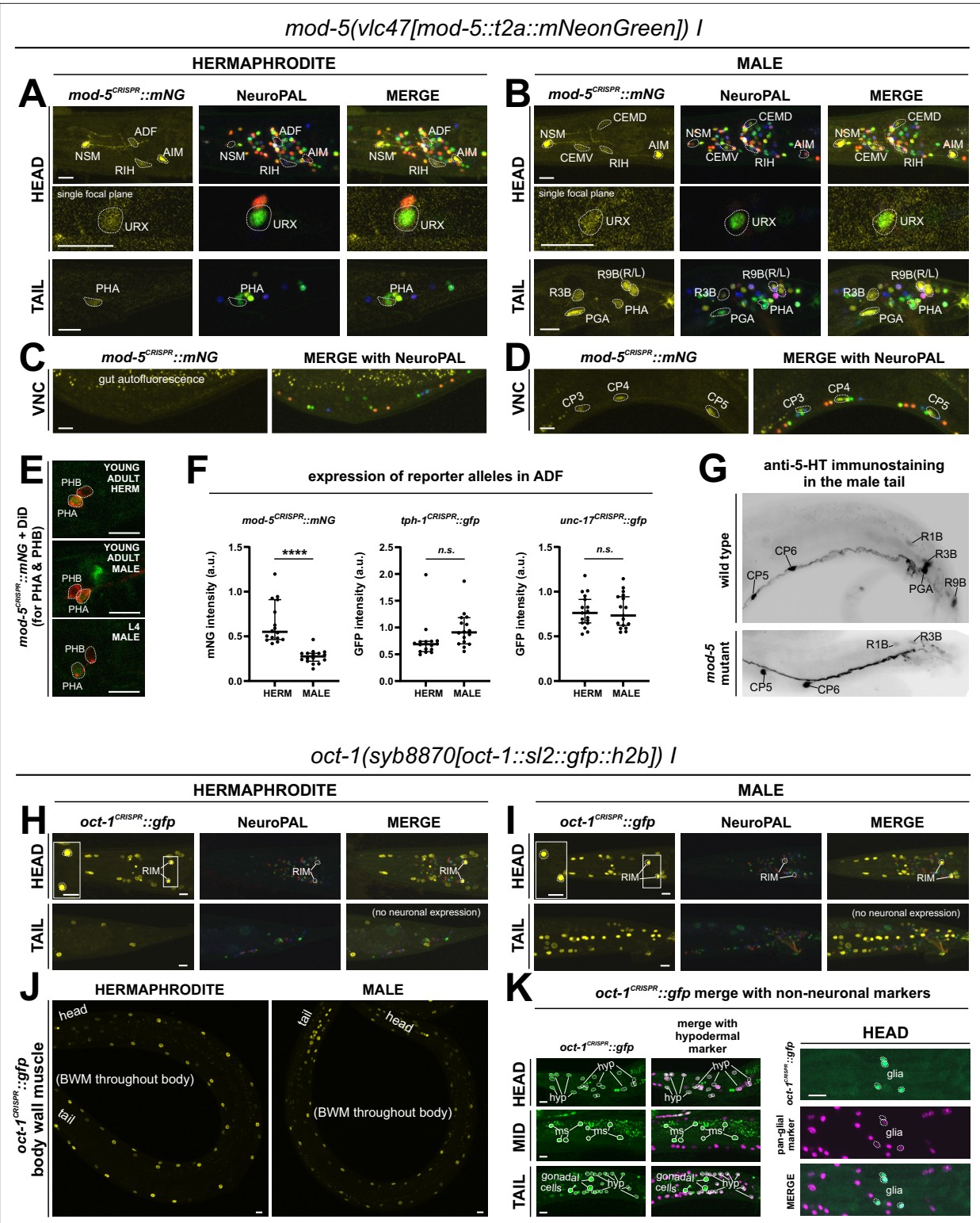

**Figure 8.** Expression of *mod-5/SERT* and *oct-1/OCT* reporter alleles in adult animals. Neuronal expression was characterized with landmark strain NeuroPAL (*otIs669*) and DiD-filling. (**A, C**) In adult hermaphrodites, *mod-5(vlc47)* is expressed in sex-shared neurons NSM, ADF, RIH, AIM, consistent with previous reports (*Jafari et al., 2011*; *Maicas et al., 2021*). In addition, we also observed expression in the phasmid neuron PHA and dim and variable expression in URX. There is no visible expression in the ventral nerve cord (VNC). (**B, D**) In adult males, *mod-5(vlc47)* is visibly expressed in NSM, RIH, AIM, as well as the male-specific neurons CEM, PGA, R3B, R9B, and CP1 to CP6. Expression in ADF is often not detected (see F). (**E**) DiD-filling confirms *mod-5(vlc47)* expression in phasmid neuron class PHA, and not PHB, in young adults in both sexes (L4 male image is to facilitate neuron ID in

*Figure 8 continued on next page*

*Figure 8 continued*

adults, because the positions of the two neuron classes can change in males during the L4 to adult transition). (**F**) Expression of *mod-5(vlc47)* in ADF is stronger in hermaphrodites than in males. Each dot represents a single animal. Expression is not sexually dimorphic for the reporter alleles of either the serotonin-synthesizing enzyme *tph-1* or the vesicular acetylcholine transporter *unc-17*. Expression was normalized against expression in other reporter-expressing neurons. Statistics, Mann-Whitney test. (**G**) In the tail region of wild type males, male-specific neurons PGA, R1B, R3B, and R9B are stained positive for serotonin. In a *mod-5(n3314)* mutant background, staining is completely lost in PGA (41/41 stained animals) and significantly affected for R9B (completely lost in 31/41 animals and much dimmer in the rest), while it remains in all 41 stained animals for R1B and R3B. The staining for CP1 to CP6 are also not affected in *mod-5* mutant animals (remaining in 41/41 stained animals; image showing CP5 and CP6). (**H, I**) In adult animals, *oct-1(syb8870)* is expressed in the tyraminergic neuron class RIM in both sexes. Expression is not observed in any other neurons. (**J, K**) Outside the nervous system, *oct-1(syb8870)* is expressed in body wall muscle (BWM) throughout the worm (**J**) as well as hypodermal cells and selected head glia (**K**). Expression is also observed in gonadal cells in the male vas deferens (**K**). A pan-glial reporter *otIs870[mir-228p::3xnls::TagRFP]* and a *dpy-7p::mCherry reporter stIs10166 [dpy-7p::his-24::mCherry+unc-119(+)]* were used for glial and hypodermal identification, respectively. Scale bars, 10 μm.

release via CAT-1/VMAT. We considered the putative neurotransmitter betaine as a possible candidate, since CAT-1/VMAT is also able to package betaine (*Peden et al., 2013*; *Hardege et al., 2022*). Betaine is synthesized endogenously, within the nervous system mostly in the *cat-1/VMAT*-positive RIM neuron (*Hardege et al., 2022*), but it is also available in the bacterial diet of *C. elegans* (*Peden et al., 2013*). In vertebrates, dietary betaine is taken up by the betaine transporter BGT1 (aka SLC6A12). To test whether *cat-1/VMAT*-positive neurons may acquire betaine via BGT1-mediated uptake, we CRISPR/Cas9-engineered a reporter allele for *snf-3/BGT1*, *syb7290*. We detected expression in the betaine-synthesizing (and also tyraminergic) RIM neuron (*Figure 9*, *Tables 1 and 2*). In addition, *snf-3* is indeed expressed in all the four *cat-1/VMAT*-positive neuron classes that do not synthesize a previously known monoaminergic transmitter (CAN, AUA, and variably, RIR and ASI) (*Figure 9A and B*). These neurons may therefore take up betaine and synaptically release it via CAT-1/VMAT. The *snf-3(syb7290)* reporter allele is also expressed in the serotonergic neuron NSM (albeit variably) (*Tables 1 and 2*), thus NSM could also be a betaine uptake neuron. In addition, we also detected *snf-3(syb7290)* expression in several other neurons that do not express *cat-1(syb6486)* (*Supplementary file 1*). Expression was also observed in a substantial number of non-neuronal cell types (*Figure 9E–G*, *Table 2*, *Supplementary file 1*). These neurons and non-neuronal cells may serve to clear betaine (see Discussion, Neurotransmitter synthesis versus uptake). *snf-3(syb7290)* is not expressed in the inner and outer labial neuron classes as previously suggested (*Peden et al., 2013*); these cells were likely misidentified in the previous study and are in fact inner and outer labial glial cells (as discussed further below).

Together with the expression pattern of the uptake transporters, all *cat-1/VMAT*-positive neurons in the hermaphrodite can be matched with an aminergic neurotransmitter. We nevertheless wondered whether another presently unknown monoaminergic transmitter, e.g., histamine or other trace amine, could be synthesized by a previously uncharacterized AAAD enzyme encoded in the *C. elegans* genome, *hdl-1* (*Figure 1—figure supplement 1A*; *Hare and Loer, 2004*). We CRISPR/Cas9-engineered an *hdl-1* reporter allele, *syb1048*, but detected no expression of this reporter in the animal (*Figure 1—figure supplement 1C and D*). Attempts to amplify weak expression signals by insertion of Cre recombinase into the locus failed [*hdl-1(syb4208)*] (see Materials and methods). CeNGEN scRNA data also shows no strong transcript expression in the hermaphrodite nervous system and only detected notable expression in sperm (*Taylor et al., 2021*).

## Reporter alleles and NeuroPAL-facilitated neuron class-identification reveal novel expression patterns of neurotransmitters in the male-specific nervous system

No comprehensive scRNA atlas has yet been reported for the nervous system of the male. Based on the expression of fosmid-based reporters, we had previously assembled a neurotransmitter atlas of the *C. elegans* male nervous system in which individual neuron classes are notoriously difficult to identify (*Serrano-Saiz et al., 2017b*). We have since established a NeuroPAL landmark strain that permits more reliable identification of gene expression patterns in both the hermaphrodite and male-specific nervous system (*Tekieli et al., 2021*; *Yemini et al., 2021*). We used NeuroPAL to facilitate the analysis of the expression profiles of our CRISPR/Cas9-engineered reporter alleles in the male, resulting in updated expression profiles for 11 of the 16 knock-in reporter alleles analyzed. As in the hermaphrodite, reasons for these updates vary. In addition to the improved accuracy of neuron identification

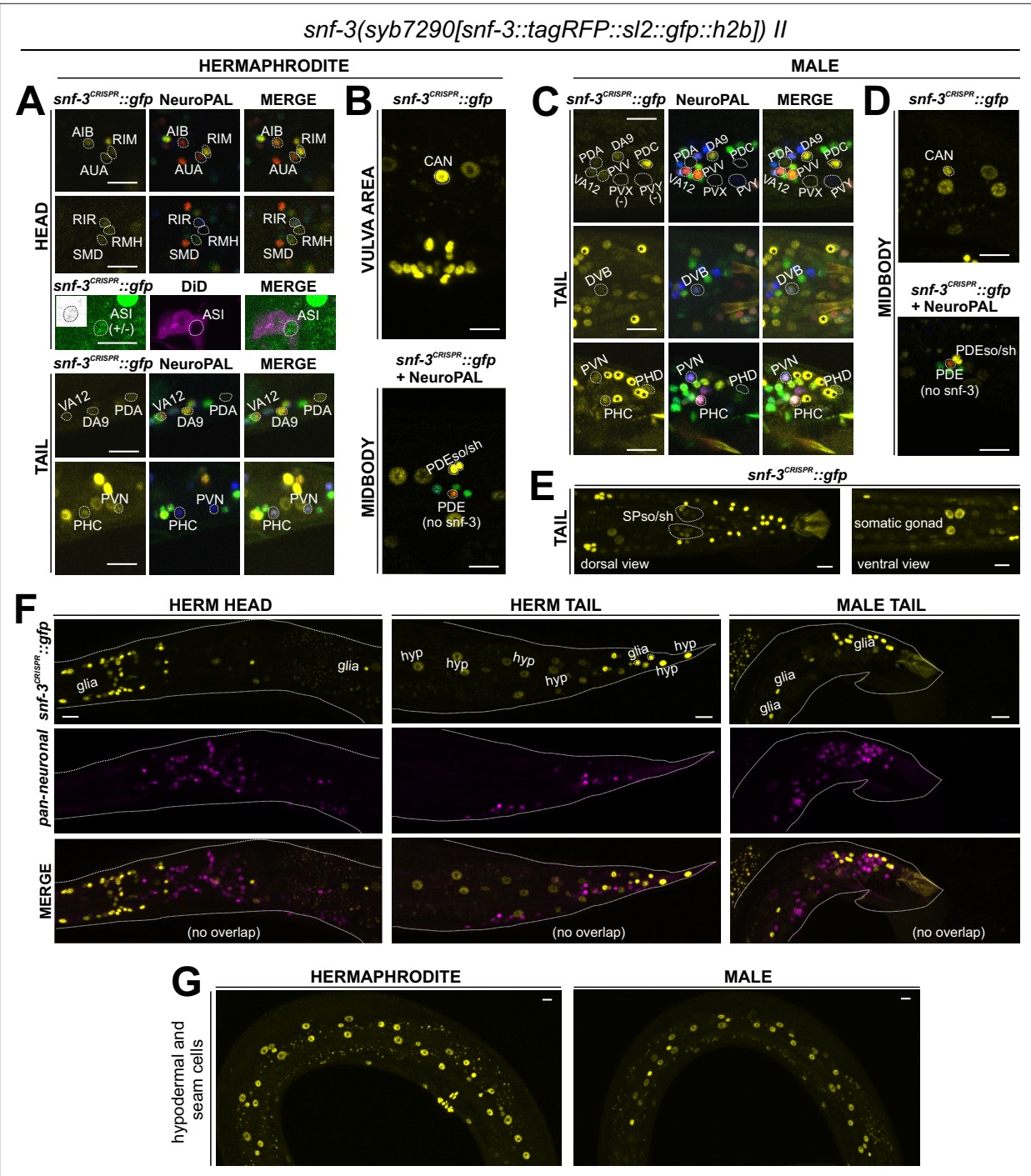

**Figure 9.** Expression of *snf-3/BGT1/SLC6A12* in adult animals. Neuronal expression was characterized with landmark strain NeuroPAL (*otIs669*) and DiD-filling. (**A, B**) In the adult hermaphrodite, neuronal expression of *snf-3(syb7290)* is detected in *cat-1/VMAT*-positive neurons AUA, CAN, and dimly and variably, RIR and ASI (confirmed with DiD-filling). In addition, it is also expressed in *cat-1/VMAT*-negative neurons AIB, RIM, RMH, SMD, VA12, DA9, PDA, PHC, PVN as labeled, as well as more neurons listed in ***Supplementary file 1***. In the midbody, expression is not detected in PDE (dopaminergic, *cat-1*-positive) but is in its associated glial cells. It is also detected in multiple vulval support cells (**B**) and some epithelial cells near the somatic gonad. (**C**) In the adult male, in addition to its expression in sex-shared neurons as in hermaphrodites, *snf-3(syb7290)* is also expressed in male-specific neuron class PDC, as well as in PHD and variably in PVV. (**D**) Similarly to its expression in hermaphrodites, *snf-3(syb7290)* is detected in CAN and PDE-associated glial cells, but not PDE neurons, in males. (**E**) In the male tail, *snf-3(syb7290)* is expressed in a number of glial cells including the spicule sockets and/or sheath cells (dorsal view). It is also detected in the somatic gonad (ventral view). (**F**) *snf-3(syb7290)* is broadly expressed in most if not all glia in both sexes. Glial cell type is determined by cell location and the appearance of their nuclei in Normarski. To confirm they are not neurons, a pan-neuronal marker (UPN, or 'uber pan-neuronal', a component in NeuroPAL) is used to determine non-overlapping signals between the two reporters. Head expression in

*Figure 9 continued on next page*

*Figure 9 continued*

the male is very similar to that in the hermaphrodite and thus not shown. (**G**) *snf-3(syb7290)* is broadly expressed in hypodermal and seam cells in both sexes. Scale bars, 10 μm. Asterisks, non-neuronal expression.

provided by NeuroPAL, in some cases there are true differences of expression patterns between the fosmid-based reporters and reporter alleles. We elaborate on these updates for individual reporter alleles below.

## Expression of reporter alleles of Glu/ACh/GABA markers in the male-specific nervous system

We analyzed *eat-4/VGLUT* (*syb4257*), *unc-17/VAChT* (*syb4491*), *unc-25/GAD* (*ot1372*), and *unc-47/VGAT* (*syb7566*) expression in the male-specific nervous system using NeuroPAL landmark strains (*otIs696* for *eat-4* and *otIs669* for all others)(*Figures 10 and 11*). Of all those reporter alleles, *unc-25/GAD* (*ot1372*) was the only one with no updated expression. Specifically, in addition to confirming presence of expression of the *unc-25(ot1372)* reporter allele in CP9, EF1/2, EF3/4, we also confirmed its *lack* of expression in anti-GABA-positive neurons R2A, R6A, and R9B (*Gendrel et al., 2016*; *Serrano-Saiz et al., 2017b*; *Figure 11A*, *Supplementary file 3*).

In the preanal ganglion, we observed weak expression of *unc-17(syb4491) in* DX3/4 (*Figure 10B*, *Supplementary file 3*), hence assigning previously unknown neurotransmitter identity to these neurons. Related to DX3/4, we also confirmed expression of *unc-17* in DX1/2 in the dorsorectal ganglion, consistent with fosmid-based reporter data (*Supplementary file 3*; *Serrano-Saiz et al., 2017b*). In the lumbar ganglion, we detected novel expression of *unc-17(syb4491)* in five pairs of type B ray neurons, namely R1B, R4B, R5B, R7B, and R9B (*Figure 10B*, *Supplementary file 3*). Expression in all these neurons is low, possibly explaining why it is not observed with an *unc-17* fosmid-based reporter (*Serrano-Saiz et al., 2017b*).

In the ventral nerve cord, we found additional, very weak expression of *eat-4(syb4257)* in CA1 to CA4 (*Figure 10A*, *Supplementary file 3*), as well as weak expression of *unc-17(syb4491)* in CP1 to CP4 (*Figure 10B*, *Supplementary file 3*), all undetected by previous analysis of fosmid-based reporters (*Serrano-Saiz et al., 2017b*). Conversely, two neurons lack previously reported expression of fosmid-based reporters; CP9 does not show visible *unc-17(syb4491)* expression (*Figure 10B*) and neither does CA9 show visible expression of *unc-47(syb7566)* expression (*Figure 11C*). We also realized that the neuron identifications of CA7 and CP7 were previously switched (*Serrano-Saiz et al., 2017b*), due to lack of proper markers for those two neurons. With NeuroPAL, we are now able to clearly distinguish the two and update their classic neurotransmitter reporter expression: CA7 expresses high levels of *eat-4(syb4257)* (*Figure 10A*, *Supplementary file 3*), very low levels of *unc-17(syb4491)* (*Figure 10B*), and no *unc-47(syb7566)* (*Figure 11C*); CP7 expresses no *eat-4(syb4257)* (*Figure 10A*, *Supplementary file 3*), very low levels of *unc-17(syb4491)* (*Figure 10B*), and very low levels of *unc-47(syb7566)* as well (*Figure 11C*). Taken together, the analysis of reporter alleles reveals a remarkable diversity of CA and CP neurons, summarized in *Figure 10C*.

In the head, we detected expression of *unc-47(syb7566)* in the male-specific neuron class MCM (*Figure 11B*, *Supplementary file 3*), previously not observed with fosmid-based reporters. Consistent with fosmid-based reporter data, the other male-specific head neuron class, CEM, shows expression of *unc-17(syb4491)* (*Supplementary file 3*) and *unc-47(syb7566)* (*Figure 11B*, *Supplementary file 3*) reporter alleles.

## Expression of reporter alleles for monoaminergic neurotransmitter pathway genes in the male-specific nervous system

We analyzed the expression of reporter alleles for genes involved in monoamine biosynthesis and uptake in the male-specific nervous system: *cat-1/VMAT* (*syb6486*), *tph-1/TPH* (*syb6451*), *cat-2/TH* (*syb8255*), *bas-1/AAAD* (*syb5923*), *tdc-1/TDC* (*syb7768*), *tbh-1/TBH* (*syb7786*), *mod-5/SERT* (*vlc47*), *oct-1/OCT* (*syb8870*), and *snf-3/BGT1* (*syb7290*). As in the hermaphrodite nervous system, we used the NeuroPAL reporter landmark (*otIs669*) for neuron ID (*Tekieli et al., 2021*). We found novel expression patterns in all male-specific ganglia (*Figures 12 and 13*, *Supplementary file 3*).

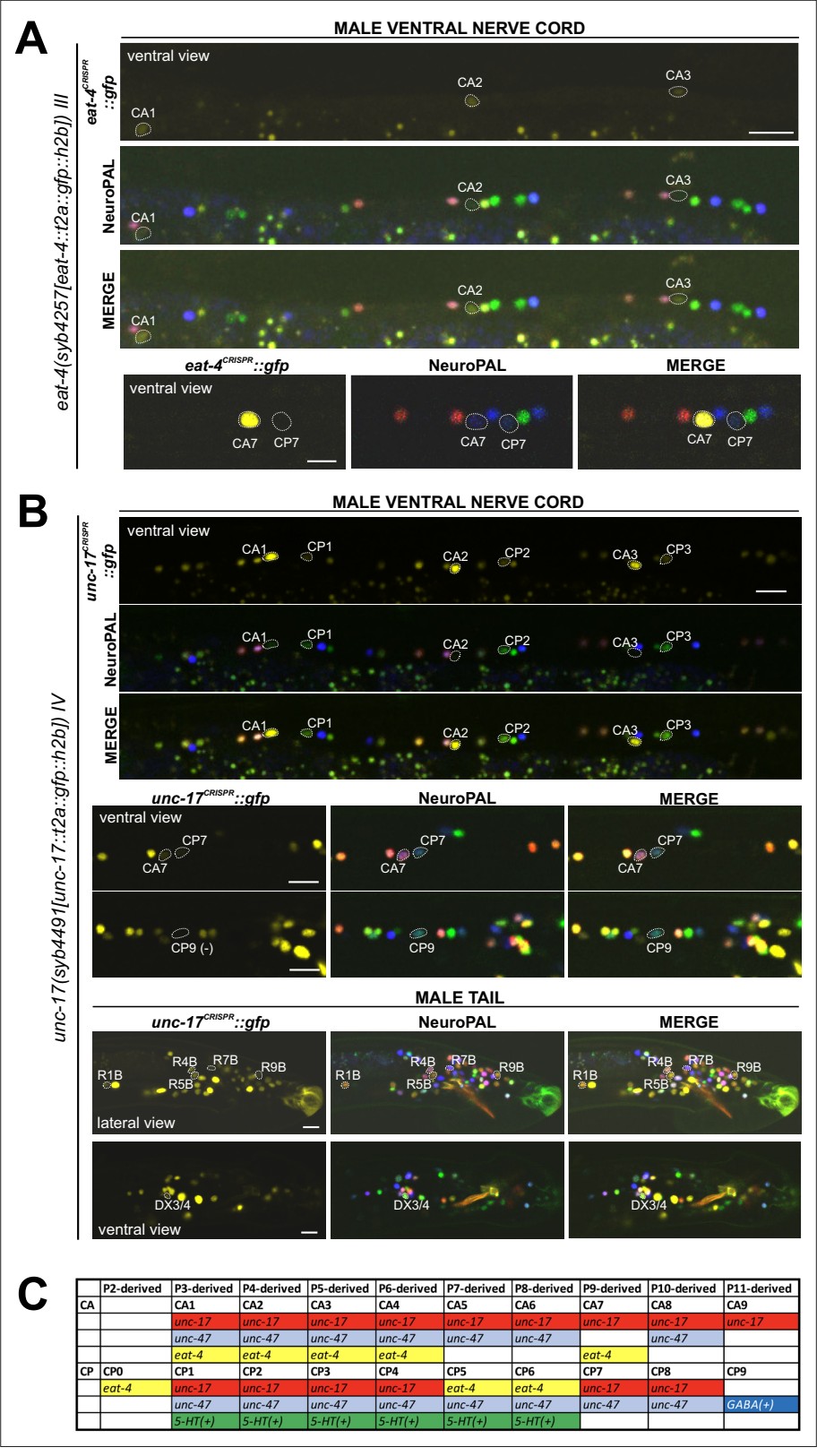

**Figure 10.** Expression of *eat-4/VGLUT* and *unc-17/VAChT* reporter alleles in the adult male. Neuronal expression of *eat-4(syb4257)* and *unc-17(syb4491)* was characterized with landmark strain NeuroPAL (*otIs696* and *otIs669*, respectively). Only selected neurons are shown to illustrate updates from previous studies. See *Supplementary file 3* for a complete list of neurons. (**A**) *eat-4(syb4257)* expression. Top, long panels: CA1, CA2, and CA3 show

*Figure 10 continued on next page*

*Figure 10 continued*

visible, albeit very dim, novel expression of *eat-4* (also expressed in CA4). Bottom panels: CA7 strongly expresses *eat-4(syb4257)*, whereas CP7 does not. Neuron IDs for these two neurons were previously switched (***Serrano-Saiz et al., 2017b***). (**B**) *unc-17(syb4491)* expression. Top, long panels: ventral view of a male ventral nerve cord showing high levels of expression in CA1, CA2, and CA3 and previously unreported low levels of expression in CP1, CP2, and CP3. Middle panels: low levels of expression in CA7 and CP7. There is no visible expression in CP9. Bottom panels: lateral view of a male tail showing previously unreported dim expression in R1B, R4B, R5B, R7B, and R9B; ventral view of the preanal ganglion showing expression in DX3/4. Scale bars, 10 μm. (**C**) The updated neurotransmitter atlas underscores the molecular diversity of the male-specific ventral cord neuron class CA and CP. Based on their expression patterns for neurotransmitter genes, these neurons can be grouped into four CA and five CP subclasses.

## Serotonin/5-HT synthesis

Serotonergic identity had been assigned to several male-specific neurons before (CP1 to CP6, R1B, R3B, R9B) (***Loer and Kenyon, 1993***), and we validated these assignments with our reporter alleles (***Figure 12***, ***Supplementary file 3***). In addition, we detected previously unreported expression of *tph-1* (***Figure 12B***) in the male-specific head neuron class CEM, as well as in a subset of B-type ray sensory neurons, R4B and R7B. However, not all of the neurons display additional, canonical serotonergic neuron features: While R4B and R7B express *bas-1(syb5923)* (with R4B expressing it variably) to generate serotonin, neither neuron was detected by anti-serotonin staining in the past. On the other hand, R9B and CEM stain positive for 5-HT (***Serrano-Saiz et al., 2017b***), but they do not express *bas-1(syb5923)*, indicating that they may be producing 5-HTP rather than 5-HT (sertonin)(see more below on serotonin uptake). In addition, R4B and R9B, but not R7B or CEM, express *cat-1(syb6486)* for vesicular release of serotonin.

In the ventral nerve cord, consistent with previous fosmid-based reporter data (***Serrano-Saiz et al., 2017b***), we observed the expression of *cat-1(syb6486)* and *tph-1(syb6451)* in CP1 to CP6 (***Supplementary file 3***). Additionally, we also detected novel expression of *bas-1(syb5923)* in CP1 to CP4 and strongly in CP5 and CP6 (***Figure 12C***, ***Supplementary file 3***). This updated expression supports the serotonergic identities of these neurons, which had been determined previously based only on their expression of *cat-1/VMAT* reporters and positive staining for serotonin (***Loer and Kenyon, 1993***; ***Serrano-Saiz et al., 2017b***).

## Dopamine synthesis

We found that the expression of the dopamine-synthesizing *cat-2(syb8255)* reporter allele precisely matched previous assignments of dopaminergic identity (***Sulston et al., 1975***; ***Sulston et al., 1980***; ***Lints and Emmons, 1999***), i.e., expression was detected exclusively in R5A, R7A, and R9A (***Figure 13A***, ***Supplementary file 3***), in addition to all sex-shared dopaminergic neurons. All these neurons show matching expression of *bas-1/AAAD*, the other essential enzyme for dopamine synthesis, and *cat-1/VMAT*, the vesicular transporter for dopamine (***Figure 12A and C***; ***Supplementary file 3***).

## Tyramine and octopamine synthesis

Reporter alleles for the two diagnostic enzymes, *tdc-1/TDC* and *tbh-1/TBH*, confirm the previously reported assignment of HOA as tyraminergic (***Serrano-Saiz et al., 2017b***), based on the presence of *tdc-1(syb7768)* but absence of *tbh-1(syb7786)* expression (***Figure 13 B and C***). The *tdc-1* reporter allele reveals a novel site of expression in R7A. Due to lack of *tbh-1* expression, R7A therefore classifies as another tyraminergic neuron. Both HOA and R7A also co-express *cat-1/VMAT* for vesicular release of tyramine (***Figure 12A***).

We detected no neurons in addition to the sex-shared RIC neuron class that shares all features of a functional octopaminergic neuron, i.e., co-expression of *tbh-1/TBH*, *tdc-1/TDC*, and *cat-1/VMAT*. While one male-specific neuron, R8B, shows an overlap of expression of *tdc-1(syb7768)* and *tbh-1(syb7786)* (***Figure 13B and C***), indicating that these neurons can synthesize octopamine, R8B does not express *cat-1(syb6486)*, indicating that it cannot engage in vesicular release of octopamine.

Curiously, while there are no other male-specific neurons that co-express *tdc-1* and *tbh-1*, several male-specific neurons express *tbh-1*, but not *tdc-1* (***Figure 13B and C***; ***Table 2***, ***Supplementary file 3***). The absence of the TDC-1/AAAD protein, which produces tyramine, the canonical substrate of the

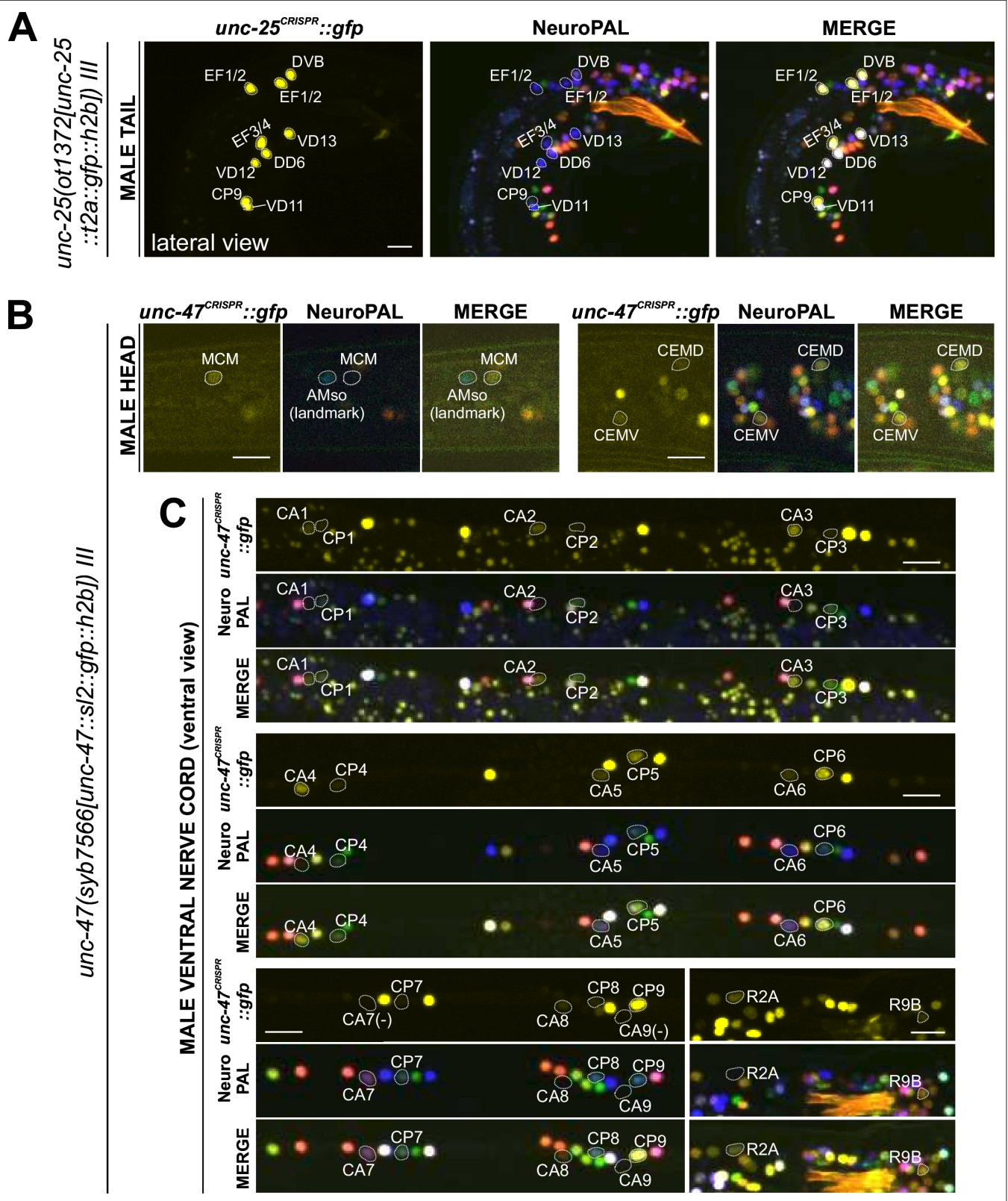

**Figure 11.** Expression of GABAergic reporter alleles in the adult male. Neuronal expression of *unc-25(ot1372)* and *unc-47(syb7566)* reporter alleles was characterized with landmark strain NeuroPAL (*otIs669*). Only selected neurons are shown to illustrate updates from previous reports. See **Supplementary file 3** for a complete list of neurons. (**A**) *unc-25(ot1372)* is expressed in male-specific CP9 and EF neurons as well as a few sex-shared neurons, all consistent with previous reports (**Gendrel et al., 2016**; **Serrano-Saiz et al., 2017b**). (**B**) *unc-47(syb7566)* shows expression in male head

*Figure 11 continued on next page*

*Figure 11 continued*

neuron classes MCM and CEM, the former previously undetected and the latter consistent with fosmid-based reporter *otIs564.* (**C**) *unc-47(syb7566)* shows expression in a number of ventral cord CA and CP neurons, largely consistent with reported *otIs564* fosmid-based reporter expression except for no visible expression of *syb7566* in CA7 (due to its initial confusion with CP7, described in **Figure 10**) and presence of very dim expression in CP7. The *syb7566* reporter allele is also not visible in CA9. Scale bars, 10 µm.

TBH-1 enzyme (**Figure 1A**), indicates that TBH-1 must be involved in the synthesis of a compound other than octopamine. Moreover, *bas-1/AAAD* is expressed in several of the *tbh-1*(+); *tdc-1*(-) neurons (R1B, R2B, R3B, R4B, and R7B) (**Figure 12C**, **Table 2**, **Supplementary file 3**). Rather than using L-Dopa or 5-HTP as substrate, BAS-1/AAAD may decarboxylate other aromatic amino acids, which then may serve as a substrate for TBH-1. We consider the trace amine phenylethanolamine (PEOH) as a candidate end product (see Discussion).

## Other monoaminergic neurons

In the preanal ganglion, we detected novel expression of the *cat-1(syb6486)* reporter allele in the cholinergic PDC, PVX, and PVY neurons (**Figure 12A**). Intriguingly, just as the sex-shared neuron AVL (**Figure 6C**), these neurons express no other serotonergic, dopaminergic, tyraminergic, or octopaminergic pathway genes. However, we did find PDC (but not PVX or PVY) to express the betaine uptake transporter reporter allele *snf-3(syb7290)* (**Figure 9**; more below). PVX and PVY may synthesize or uptake another aminergic transmitter. Such presumptive transmitter is not likely to be synthesized by *hdl-1/AAAD* since we detected no expression of the *hdl-1* reporter allele *syb4208* in the male nervous system.

The expression pattern of the *bas-1/AAAD,* which had not been previously analyzed in the male-specific nervous system, reveals additional novelties. In addition to the 'canonical' serotonergic and dopaminergic neurons described above, we detected *bas-1(syb5923)* reporter allele expression in a substantial number of additional neurons, including the tyraminergic HOA and R7A neurons, but also the DVE, DVF, R2A, R3A, R6A, R8A, R2B, R6B, R7B, PCB, and SPC neurons (**Figure 12C**, **Supplementary file 3**). As described above, a subset of the neurons co-express *tbh-1(syb7786)* (most B-type ray neurons), a few co-express *tdc-1(syb7768)* (HOA and several A-type ray neurons), and several co-express neither of these two genes. Only a subset of these neurons express *cat-1(syb6486)*. Taken together, this expression pattern analysis argues for the existence of additional monoaminergic signaling system(s) (**Table 2**).

## Serotonin/5-HT uptake

In the male-specific nervous system, we detected *mod-5/SERT* reporter allele expression in CEM, PGA, R3B, R9B, and ventral cord neurons CP1 to CP6 (**Figure 8D**). We found that anti-serotonin staining in CP1 to CP6, R1B, and R3B is unaffected in *mod-5(n3314)* mutant animals, consistent with these neurons expressing the complete serotonin synthesis machinery (i.e. *tph-1* and *bas-1*) (**Table 2**, **Figure 8B, D, and G**; **Supplementary file 3**). Hence, like several other monoaminergic neurons, these serotonergic neurons both synthesize, synaptically release, and reuptake serotonin. In contrast, anti-serotonin staining is lost from the R9B and PGA neurons of *mod-5(n3314)* mutant animals, indicating that the presence of serotonin in these neurons depends on serotonin uptake, consistent with them not expressing the complete serotonin synthesis pathway (**Table 2**; **Supplementary file 3**). Since R9B and PGA express *cat-1/VMAT* (**Figure 12A**), these neurons have the option to utilize serotonin for vesicular release after *mod-5*-dependent uptake.

## Tyramine and betaine uptake

We did not observe *oct-1(syb8870)* reporter allele expression in male-specific neurons. As in the hermaphrodite nervous system, we detected *snf-3(syb7290)* in a number of neurons that do not express CAT-1/VMAT (**Supplementary file 1**), including in male-specific neurons PHD, and variably, PVV (**Figure 9C**). As mentioned earlier, the male-specific neuron PDC expresses both *cat-1(syb6486)* and *snf-3(syb7290)*, making it a likely betaine-signaling neuron.

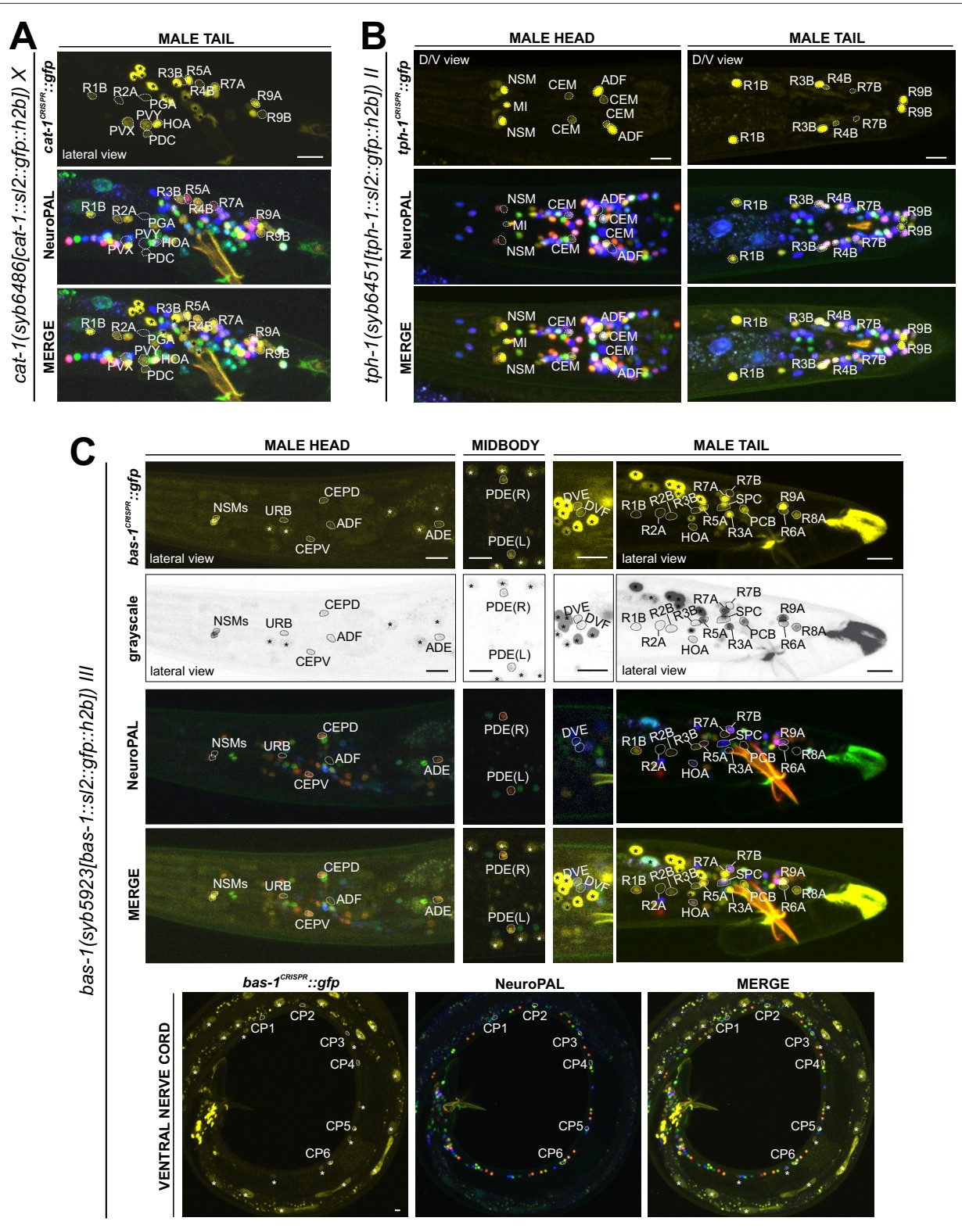

**Figure 12.** Expression of the *cat-1/VMAT*, *tph-1/TPH*, and *bas-1/AAAD* reporter alleles in the adult male. Neuronal expression was characterized with landmark strain NeuroPAL (*otIs669*). (**A**) Novel *cat-1(syb6486)* expression is seen in male-specific neurons PDC, PVY, PVX, R2A, and R4B. Consistent with previous reports, it is also expressed in HOA, PGA, R5A, R7A, R9A, R1B, and R8B. Its expression in ventral cord neurons CP1 to CP6 is consistent with earlier studies. (**B**) *tph-1(syb6451)* is expressed in male-specific head neuron class CEM and sex-shared neurons ADF, NSM, and MI. Similar to its expression in hermaphrodites, *tph-1* in MI was previously undetected. In the tail, in addition to previously determined expression in R1B, R3B, and R9B,

*Figure 12 continued on next page*

*Figure 12 continued*

*tph-1(syb6451)* is also expressed at very low levels in R4B and R7B. Ventral cord expression of *tph-1(syb6451)* in CP1 to CP6 is consistent with previous reports and thus not shown here. (**C**) *bas-1(syb5923)* is expressed in previously identified NSM, ADE, PDE, and CEP neurons. In addition, we detected weak expression in URB as in the hermaphrodite. We also updated *bas-1/AAAD* expression in 39 male-specific neurons (see **Supplementary file 3** for complete list). Neurons are also shown in grayscale for clearer visualization in some cases. Scale bars, 10 μm. Asterisks, non-neuronal expression, also see **Figure 14** and **Figure 14—figure supplement 1**.

## Sexually dimorphic neurotransmitter deployment in sex-shared neurons

### eat-4/VGLUT

We had previously noted that a fosmid-based *eat-4/VGLUT* reporter is upregulated in the sex-shared neuron PVN, specifically in males (**Serrano-Saiz et al., 2017b**). Since PVN is also cholinergic (**Figure 4D**; **Pereira et al., 2015**), this observation indicates a sexually dimorphic co-transmission configuration. As described above (**Figure 4B**, **Supplementary file 2**), our *eat-4* reporter allele revealed low levels of *eat-4/VGLUT* expression in hermaphrodites PVN, but in males the *eat-4* reporter allele showed strongly increased expression, compared to hermaphrodites. Hence, rather than being an 'on' vs. 'off' dimorphism, dimorphic *eat-4/VGLUT* expression in male PVN resembles the 'scaling' phenomenon we had described previously for *eat-4/VGLUT* in male PHC neurons, compared to hermaphrodite PHC neurons (**Serrano-Saiz et al., 2017a**). Both PHC and PVN display a substantial increase in the amount of synaptic output of these neurons in males compared to hermaphrodites (**Cook et al., 2019**), providing a likely explanation for such scaling of gene expression. The scaling of *eat-4/VGLUT* expression in PVN is not accompanied by scaling of *unc-17/VAChT* expression, which remains comparable in both sexes (**Figure 4D**).

We also examined AIM, another neuron class that was previously reported to be sexually dimorphic in that AIM expresses *eat-4/VGLUT* fosmid-based reporters in juvenile stages in both sexes, whereas upon sexual maturation its neurotransmitter identity is switched from being glutamatergic to cholinergic only in adult males and not hermaphrodites (**Pereira et al., 2015**; **Pereira et al., 2019**). With the *eat-4(syb4257)* reporter allele, we also detected a downregulation of *eat-4* expression to low levels in young adult males and almost complete elimination in 2-day-old adult males, while expression in hermaphrodites stays high.

### unc-17/VAChT

The *unc-17/VAChT* reporter allele *syb4491* confirms that cholinergic identity is indeed male-specifically turned on in the AIM neurons (**Figure 4C**), thereby confirming the previously reported neurotransmitter switch (**Pereira et al., 2015**). The fosmid-based *unc-17* reporter also showed sexually dimorphic expression in the AVG neurons (**Serrano-Saiz et al., 2017b**). This is also confirmed with the *unc-17* reporter allele, which shows dim and variable expression in hermaphrodites and slightly stronger, albeit still dim, AVG expression in males (**Figure 4C**, showing a hermaphrodite representing animals with no visible expression and a male with representative dim expression).

### unc-47/VGAT

*unc-47(syb7566)* confirms previously reported sexually dimorphic expression of *unc-47/VGAT* in several sex-shared neurons, including ADF, PDB, PVN, PHC, AS10, and AS11 (**Figure 5B**, right side panels) (**Serrano-Saiz et al., 2017b**). The assignment of AS10 was not definitive in our last report (we had considered either DA7 or AS10), but with the help of NeuroPAL the AS10 assignment could be clarified. In all these cases expression was only detected in males and not hermaphrodites. It is worth mentioning that expression of the mCherry-based *unc-47/VGAT* fosmid-based reporter (*otIs564*) in some of these neurons was so dim that it could only be detected through immunostaining against the mCherry fluorophore and not readily visible with the fosmid-based reporter by itself (**Serrano-Saiz et al., 2017b**; **Supplementary file 1**). In contrast, the *unc-47/VGAT* reporter allele is detected in all cases except the PQR neuron class.

### mod-5/SERT

Expression of the *mod-5(vlc47)* reporter allele is sexually dimorphic in the pheromone-sensing ADF neurons, with higher levels in hermaphrodites compared to males (**Figure 8F**). Notably, the

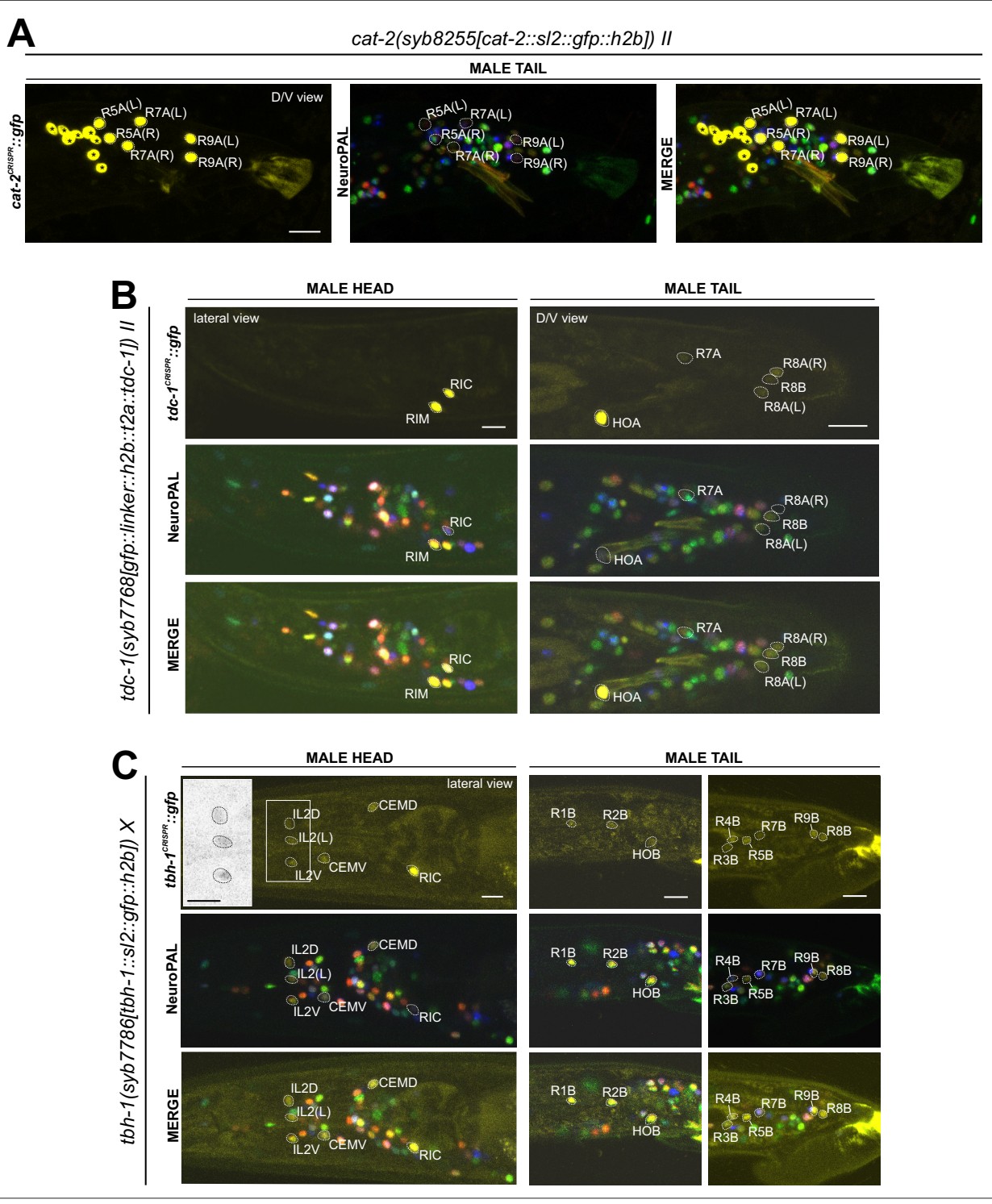

**Figure 13.** Expression of *cat-2/TH*, *tdc-1/TDC,* and *tbh-1/TBH* reporter alleles in the adult male. Neuronal expression was characterized with landmark strain NeuroPAL (*otIs669*). (**A**) *cat-2(syb8255)* is expressed in male-specific neurons R4A, R7A, and R9B. This expression, as well as its expression in sex-shared neurons PDE, CEP, and ADE, is consistent with previous reports (*Sulston et al., 1975*; *Sulston et al., 1980*; *Lints and Emmons, 1999*). (**B**) *tdc-1(syb7768)* is expressed in sex-shared neurons RIM and RIC and male-specific neurons HOA, R8A, and R8B, all consistent with previous studies (*Serrano-Saiz et al., 2017b*). We also detected weak expression in R7A. (**C**) *tbh-1(syb7786)* is expressed in RIC, consistent with its previously reported expression in hermaphrodites. As in hermaphrodites, we also detected *tbh-1(syb7786)* in IL2 neurons of the male. In male-specific neurons, previously unreported expression is detected in CEM, HOB, and all type B ray neurons except for R6B. Intriguingly, this expression pattern resembles that of *pkd-2* and *lov-1*, both genes essential for male mating functions (*Barr and Sternberg, 1999*; *Barr et al., 2001*). Inset, grayscale image showing dim expression for IL2 neurons. Scale bars, 10 μm. Asterisks, non-neuronal expression, also see *Figure 14* and *Figure 14—figure supplement 1*.

serotonin-synthesizing enzyme (*tph-1*) and vesicular acetylcholine transporter (*unc-17*) do not exhibit this dimorphism in ADF (*Figure 8F*). This suggests that the sex difference specifically involves serotonin signaling mechanisms, particularly serotonin uptake rather than synthesis.

We had previously reported that the PVW neuron stains with anti-serotonin antibodies exclusively in males but we did not detect expression of a fosmid-based reporter for the serotonin-synthesizing enzyme TPH-1 (*Serrano-Saiz et al., 2017b*). We confirmed the lack of *tph-1* expression with our new *tph-1* reporter allele in both males and hermaphrodites, and also found that hermaphrodite and male PVW does not express the reporter allele for the other enzyme in the serotonin synthesis pathway, *bas-1*. Because of very dim *cat-1::mCherry* fosmid-based reporter expression that was only detected upon anti-mCherry antibody staining, we had assigned PVW as a serotonin-releasing neuron (*Serrano-Saiz et al., 2017b*). However, we failed to detect expression of our new *cat-1/VMAT* reporter allele in PVW. Neither did we detect expression of the *mod-5(vlc47)* reporter allele. Taken together, PVW either synthesizes or uptakes serotonin by unconventional means, akin to the pharyngeal I5 neuron.

In conclusion, although there are some updates in the levels of dimorphic gene expression (PVN and ADF neuron classes), our analysis with reporter alleles does not reveal pervasive novel sexual dimorphism in sex-shared neurons compared to those that we previously identified in *Serrano-Saiz et al., 2017b*. These sexual dimorphisms are summarized in *Supplementary file 4*.

## Neurotransmitter pathway genes in glia

In vertebrates, glia can produce various signaling molecules, including neurotransmitters (*Araque et al., 2014*; *Savtchouk and Volterra, 2018*). There is some limited evidence for neurotransmitter synthesis in *C. elegans* glia. In males, it had been reported that the socket glia of spicule neurons synthesize and utilize dopamine, based on their expression of *cat-2/TH* and *bas-1/AAAD* (*Lints and Emmons, 1999*; *Hare and Loer, 2004*; *LeBoeuf et al., 2014*). We confirmed this notion with *cat-2/TH* and *bas-1/AAAD* reporter alleles (*Figure 14A*). Additionally, we detected expression of the *cat-1/VMAT* reporter allele in these cells (*Figure 14A*), indicating that these glia secrete dopamine by canonical vesicular transport. We did not detect *cat-1/VMAT* in other glial cell types. In addition to the spicule socket glia, we also observed *bas-1(syb5923)* reporter allele expression in cells that are likely to be the spicule sheath glia (*Figure 14A*), as well as in additional glial cell types in the head and tail (*Figure 14B*). We detected no glial expression of other monoaminergic synthesis machinery.

We detected no expression of vesicular transporters or biosynthetic synthesis machinery for non-aminergic transmitters in glia of either sex. This observation contrasts previous reports on GABA synthesis and release from the AMsh glial cell type (*Duan et al., 2020*; *Fernandez-Abascal et al., 2022*). We were not able to detect signals in AMsh with anti-GABA staining, nor with an SL2- or T2A-based GFP-based reporter allele for any *unc-25* isoform (*Gendrel et al., 2016*) (M Gendrel, pers. comm.; this paper).

There is, however, abundant evidence for neurotransmitter uptake by *C. elegans* glial cells, mirroring this specific function of vertebrate glia (*Henn and Hamberger, 1971*). We had previously shown that one specific glia-like cell type in *C. elegans*, the GLR glia, take up GABA via the GABA uptake transporter SNF-11 (*Gendrel et al., 2016*). We did not detect *unc-47/VGAT* fosmid-based reporter expression in the GLRs (*Gendrel et al., 2016*) and also detected no expression with our *unc-47/VGAT* reporter allele. Hence, these glia are unlikely to release GABA via classic vesicular machinery. Other release mechanisms for GABA can of course not be excluded. Aside from the *snf-11* expression in GLR glia (*Gendrel et al., 2016*), we detected expression of the putative tyramine uptake transporter *oct-1/OCT* in a number of head glial cells (*Figure 8K*), as well as broad glial expression of the betaine uptake transporter *snf-3/BGT1* in the head, midbody, and tail (*Figure 9E and F*). These results indicate tyramine and betaine clearance roles for glia.

## Neurotransmitter pathway gene expression outside the nervous system

We detected expression of a few neurotransmitter pathway genes in cells outside the nervous system. The most prominent sites of reporter allele expression are located within the gonad. We detected expression of *tdc-1(syb7768)* and *tbh-1(syb7786)* reporter alleles in the gonad of hermaphrodite as well as *tdc-1(syb7768)* expression in the neuroendocrine uv1 cells (*Figure 14C*; *Figure 14—figure supplement 1*), as previously reported (*Alkema et al., 2005*). Intriguingly, while *cat-1(syb6486)*

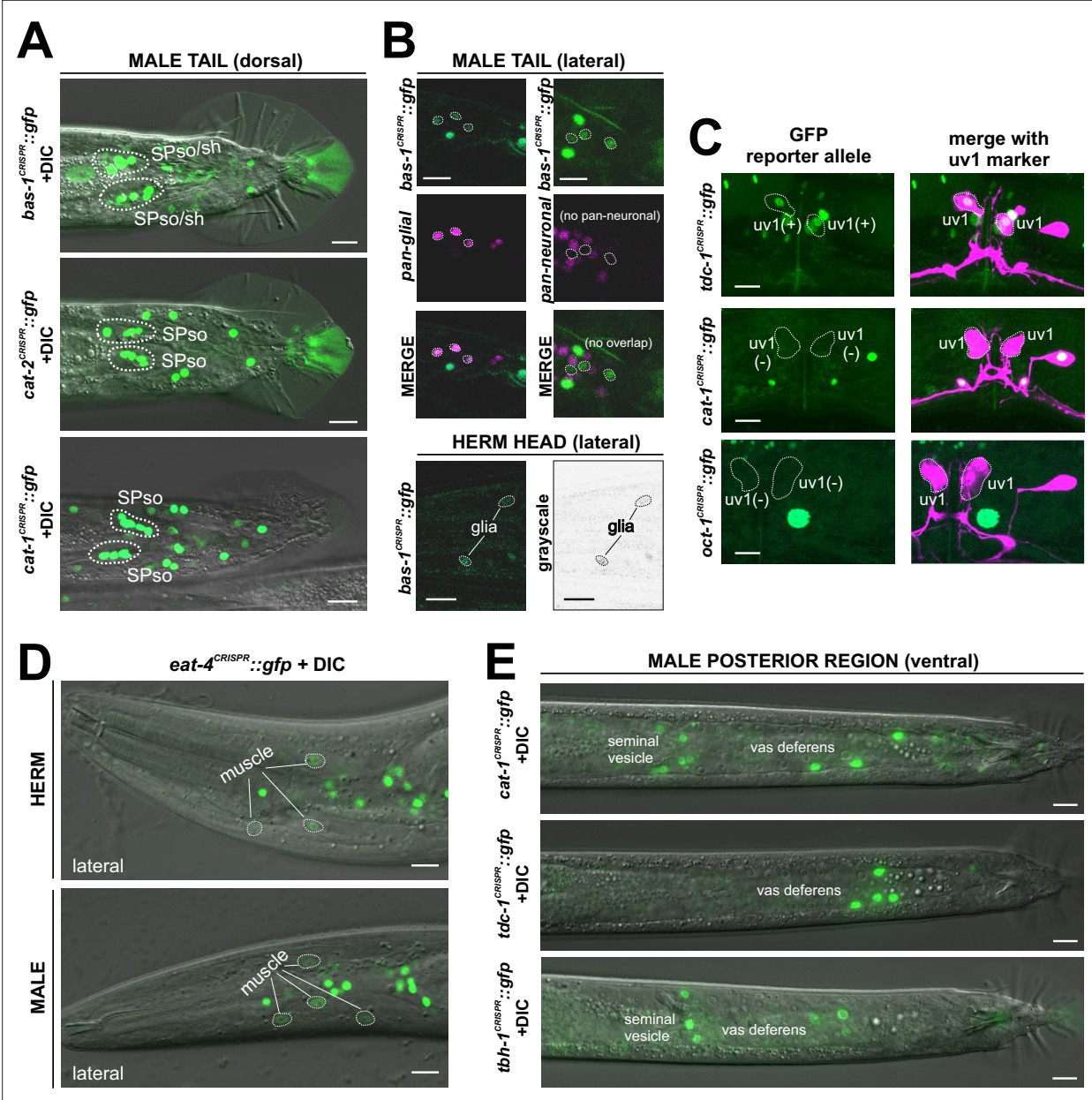

**Figure 14.** Expression of neurotransmitter pathway genes in non-neuronal cell types. Multiple neurotransmitter pathway genes show expression in glial cells (**A, B**) and other non-neuronal cell types (**C–E**). Also see *Figure 14—figure supplement 1* for whole-worm views that capture more non-neuronal expression. (**A**) *bas-1(syb5923)*, *cat-2(syb8255)*, and *cat-1(syb6486)* reporter alleles exhibit expression in the male spicule glial cell types, largely consistent with previous reports (*Lints and Emmons, 1999*; *Hare and Loer, 2004*; *LeBoeuf et al., 2014*). (**B**) Top 6 panels: *bas-1(syb5923)* is expressed in additional, multiple glial cell types in the male tail. Left 3 panels: *bas-1(syb5923)* crossed into a pan-glial reporter *otIs870[mir-228p::3xnls::TagRFP]*, confirming its expression in glial cells; right 3 panels: *bas-1(syb5923)* shows no overlap with the pan-neuronal marker component in NeuroPAL (*otIs669*). Bottom 2 panels: *bas-1(syb5923)* also shows expression in at least two glial cells in the head. A hermaphrodite head is shown here. Expression is similar in the male. (**C**) In the hermaphrodite vulval region, *tdc-1(syb7768)* is expressed in uv1, consistent with previous reports (*Alkema et al., 2005*). This expression in uv1 is not observed for either *cat-1(syb6486)* or *oct-1(syb8870)*. An *ida-1p::mCherry* integrant *vsIs269[ida-1::mCherry]* was used for identifying uv1. (**D**) Detection of *eat-4(syb4257)* expression in muscle cells in both sexes, most prominently in the head. (**E**) *cat-1(syb6486)*, *tdc-1(syb7768)*, and *tbh-1(syb7786)* are expressed in the male somatic gonad. All three have expression in the vas deferens; additionally, *cat-1* and *tbh-1* are also expressed in the seminal vesicle.

The online version of this article includes the following figure supplement(s) for figure 14:

**Figure supplement 1.** Whole-worm images showing monoaminergic pathway gene expression in different tissue types.

is expressed in a midbody gonadal cell posterior to the vulva, likely the distal valve (*Figure 6C*, *Figure 14—figure supplement 1*), we observed no expression of *cat-1(syb6486)* in the gonad or the uv1 cells (*Figure 14C*). This suggests alternative release mechanisms for tyramine and octopamine. A vertebrate homolog of the putative tyramine uptake transporter, *oct-1,* has been found to be located presynaptically and to co-purify with synaptosomes (*Berry et al., 2016*; *Matsui et al., 2016*), therefore indicating that this transporter may have the potential to also act in tyramine release, at least in vertebrate cells. However, we observed no expression of our *oct-1* reporter allele in uv1 or gonadal cells.

In the male, *tdc-1(syb7768)*, *tbh-1(syb7786)*, *cat-1(syb6486)*, and *oct-1(syb8870)* animals also show reporter expression in the somatic gonad: while all four genes are expressed in the vas deferens, *cat-1* and *tbh-1*, but not *tdc-1* or *oct-1*, are expressed in the seminal vesicle (*Figure 14C*, *Figure 8K*). A similar pattern of *cat-1(+)*; *tbh-1(+)*; *tdc-1(-)*; *oct-1(-)* is detected in several male-specific neurons and may indicate the usage of a novel transmitter (e.g. PEOH, see Discussion) by these cells. *snf-3/BGT1* is also expressed in male somatic gonad cells, indicating that these cells could also use betaine for signaling (*Figure 9E*).

The AAADs *tdc-1* and *bas-1* are also prominently expressed in the intestine, where *bas-1* has been shown to be involved in generating serotonin-derived glucosides (*Yu et al., 2023*). *bas-1*, but not *tdc-1*, is also expressed in the hypodermis and seam cells, as is the betaine uptake transporter *snf-3* (*Figure 9*, *Figure 14—figure supplement 1*). The *tph-1* reporter allele expresses in a subset of pharyngeal non-neuronal cells during the L1 to L4 larval stages of development (*Figure 6—figure supplement 1*), which is consistent with low levels of *tph-1* transcripts detected in pharyngeal muscles in the CeNGEN scRNA dataset (*Taylor et al., 2021*). Additionally, we observed previously uncharacterized *eat-4/VGLUT* expression in muscle cells in both sexes (*Figure 14D*).

## Discussion

Using CRISPR/Cas9-engineered reporter alleles we have refined and extended neurotransmitter assignments throughout all cells of the *C. elegans* male and hermaphrodite. We conclude that in both hermaphrodites and males, about one quarter of neurons are glutamatergic (*eat-4/VGLUT*-positive), a little more than half are cholinergic (*unc-17/VAChT*-positive), around 10% are GABAergic (*unc-25/GAD*-positive), and about another 10% are monoaminergic (*cat-1/VMAT*-positive). We compiled comprehensive lists for gene expression and neuron identities, which are provided in *Supplementary file 2* for hermaphrodites and *Supplementary file 3* for males. *Figure 3* presents a summary of neurotransmitter usage and atlases showing neuron positions in worm schematics. Additionally, we summarize our rationale for assigning neurotransmitter usage and updates to previously reported data in *Tables 1 and 2*, and *Supplementary file 5*. Given the complexity and nuances in determining neurotransmitter usage, we refer the reader to all the individual tables for a comprehensive description of the subject matter, rather than encouraging sole reliance on the summary in *Figure 3*.

### Neurotransmitter synthesis versus uptake

Direct detection of neurotransmitters through antibody staining has shown that at least two neurotransmitters, GABA and serotonin, are present in some neurons that do not express the synthesis machinery for these transmitters (*Tables 1 and 2*). Instead, these neurons acquire GABA and serotonin through uptaking them via defined uptake transporters, SNF-11/BGT1 for GABA (*Mullen et al., 2006*) and MOD-5/SERT for serotonin (*Ranganathan et al., 2001*; *Jafari et al., 2011*). A combination of CeNGEN scRNA transcriptome and our reporter allele data corroborates the absence of synthesis machinery in these presumptive uptake neurons (*Tables 1 and 2*). One interesting question that relates to these uptake neurons is whether they serve as 'sinks' for clearance of a neurotransmitter or whether the taken-up neurotransmitter is subsequently 'recycled' for synaptic release via a vesicular transporter. Previous data, as well as our updated expression profiles, provide evidence for both scenarios: ALA and AVF do not synthesize GABA via UNC-25/GAD, but they stain with anti-GABA antibodies in a manner that is dependent on the uptake transporter SNF-11 (*Gendrel et al., 2016*). ALA expresses *unc-47*, hence it is likely to synaptically release GABA, but AVF does not, and it is therefore apparently involved only in GABA clearance. Similarly, RIH, AIM, and PGA express the serotonin uptake transporter *mod-5/SERT* and stain for serotonin in a MOD-5-dependent manner

(*Jafari et al., 2011*) (this study), but only RIH and PGA, not AIM, expresses the vesicular transporter *cat-1/VMAT*, suggesting RIH and PGA are likely serotonergic signaling neurons whereas AIM is a clearance neuron.

Some neurons do not obviously fall into the synthesis or uptake category, most notably, the anti-GABA-antibody-positive AVA and AVB neurons (both of which conventional cholinergic neurons). None of these neurons express *unc-25/GAD*, nor the *snf-11/BGT1* uptake transporter, yet *unc-25/GAD* is required for their anti-GABA-positive staining (*Gendrel et al., 2016*). This suggests that GABA may be acquired by these neurons through non-canonical uptake or synthesis mechanisms. Also, the AVA and AVB neurons do not express UNC-47 (*Gendrel et al., 2016*; *Taylor et al., 2021*) (this study); hence, it is not clear if or how GABA is released from them. A member of the bestrophin family of ion channels has been shown to mediate GABA release from astrocyte glia in vertebrates (*Lee et al., 2010*) and, more recently, from *C. elegans* glia (*Cheng et al., 2024*; *Graziano et al., 2024*). However, while there are more than 20 bestrophin channels encoded in the *C. elegans* genome (*Hobert, 2013*), they do not appear to be expressed in the AVA or AVB neurons, based on CeNGEN scRNA data (*Taylor et al., 2021*).

The co-expression of a specific uptake transporter and a vesicular transporter corroborates the potential usage of betaine as a neurotransmitter. Betaine is known to be synthesized in *C. elegans* but is also taken up via its diet (*Peden et al., 2013*; *Hardege et al., 2022*). Betaine has documented effects on *C. elegans* behavior and acts via activation of several betaine-gated ion channels (*Peden et al., 2013*; *Hardege et al., 2022*). Expression of biosynthetic enzymes suggests betaine production in at least the RIM neuron class, which also expresses the vesicular transporter *cat-1/VMAT*, capable of transporting betaine (*Hardege et al., 2022*). The expression of the betaine uptake transporter *snf-3/BGT1* in CAN, AUA, RIR, ASI, and male-specific neuron PDC, coupled with their co-expression of *cat-1/VMAT*, suggests that several distinct neuron classes in different parts of the nervous system may uptake betaine and engage in vesicular betaine release via CAT-1/VMAT to gate betaine-activated ion channels, such as ACR-23 (*Peden et al., 2013*) or LGC-41 (*Hardege et al., 2022*). Additionally, we detected the *snf-3/BGT1* reporter allele in several other neuron classes that do not co-express *cat-1/VMAT*. This indicates that these neurons could function as betaine clearance neurons.

Lastly, based on sequence similarity and expression pattern, we predict that the ortholog of the OCT subclass of SLC22 family, *oct-1*, could serve as a tyramine uptake transporter in *C. elegans*. We identified RIM as the only neuron expressing an *oct-1* reporter allele, suggesting that like several other monoaminergic neuron classes, RIM both synthesizes its monoaminergic transmitter and reuptakes it after release.

## Evidence for usage of currently unknown neurotransmitters
### Novel amino acid transmitters?
*unc-47/VGAT* is expressed in a substantial number of non-GABAergic neurons (95 out of 302 total neurons in hermaphrodites, plus 61 out of 93 male-specific neurons). However, expression in many of these non-GABAergic neurons is low and variable and such expression may not lead to sufficient amounts of a functional gene product. Yet, in some neurons (e.g. the SIA neurons) expression of *unc-47* is easily detectable and robust (based on fosmid-based reporter, reporter allele, and scRNA data), indicating that VGAT may transport another presently unknown neurotransmitter (*Gendrel et al., 2016*). In vertebrates, VGAT transports both GABA and glycine, and the same is observed for UNC-47 in vitro (*Aubrey et al., 2007*). While the *C. elegans* genome encodes no easily recognizable ortholog of known ionotropic glycine receptors, it does encode anion channels that are closely related by primary sequence (*Hobert, 2013*). Moreover, a recently identified metabotropic glycine receptor, GPR158 (*Laboute et al., 2023*), has a clear sequence ortholog in *C. elegans, F39B2.8*. Therefore, glycine may also act as a neurotransmitter in *C. elegans*. VGAT has also been shown to transport β-alanine (*Juge et al., 2013*), another potential, but as yet unexplored, neurotransmitter in *C. elegans*. However, it needs to be pointed out that most of the additional *unc-47*-positive neurons do not co-express the LAMP-type UNC-46 protein, which is important for sorting UNC-47/VGAT to synaptic vesicles in conventional GABAergic neurons (*Schuske et al., 2007*). In vertebrates, the functional UNC-46 ortholog LAMP5 is only expressed and required for VGAT transport in a subset of VGAT-positive, GABAergic neurons (*Tiveron et al., 2016*; *Koebis et al., 2019*), indicating that alternative vesicular sorting mechanisms exist for UNC-47/VGAT.

## Novel monoaminergic transmitters?

Three neuron classes (AVL, PVX, and PVY) express *cat-1/VMAT* but do not express the canonical synthesis machinery for serotonin, tyramine, octopamine, or dopamine. Neither do they show evidence for uptake of known monoamines. There are also several *cat-1/VMAT*-positive male-specific neurons that express only a subset of the biosynthetic machinery involved in the biosynthesis of known aminergic transmitters in the worm. That is, some neurons express *cat-1/VMAT* and *bas-1/AAAD*, but none of the previously known enzymes that produce the substrate for BAS-1, i.e., CAT-2 or TPH-1 (*Figure 1A*). In these neurons, BAS-1/AAAD may decarboxylate an unmodified (i.e. non-hydroxylated) aromatic amino acid as substrate to produce, for example, the trace amine PEA from phenylalanine (*Table 2*, *Figure 1—figure supplement 1A*). A subset of these neurons (all being B-type ray sensory neurons) co-express *tbh-1*, which may use PEA as a substrate to produce the trace amine, PEOH. PEOH is a purported neurotransmitter in Aplysia (*Saavedra et al., 1977*) and the vertebrate brain (*Saavedra and Axelrod, 1973*) and can indeed be detected in *C. elegans* extracts (F Schroeder, pers. comm.).

*bas-1/AAAD* may also be responsible for the synthesis of histamine, an aminergic neurotransmitter that can be found in extracts of *C. elegans* (*Pertel and Wilson, 1974*). The only other AAAD that displays reasonable sequence similarity to neurotransmitter-producing AAADs is the *hdl-1* gene (*Hare and Loer, 2004*; *Hobert, 2013*; *Figure 1—figure supplement 1B*), for which we, however, did not detect any expression in the *C. elegans* nervous system (*Figure 1—figure supplement 1C and D*). Since there are neurons that only express *bas-1/AAAD*, but no enzyme that produces canonical substrates for *bas-1/AAAD* (*tph-1/TPH, cat-2/TH*; *Figure 1A*), and since at least a subset of these neurons express the monoamine transporter *cat-1/VMAT* (*Table 2*), *bas-1/AAAD* may be involved in synthesizing another currently unknown bioactive monoamine.

Conversely, based on the expression of *tph-1*, but concurrent absence of *bas-1/AAAD,* the pharyngeal MI neuron, hermaphrodite VC4 and VC5, and male neurons CEM and R9B may produce 5-HTP (*Table 2*). 5-HTP may either be used directly as a signaling molecule or it may be metabolized into some other serotonin derivative, an interesting possibility in light of serotonin derivatives produced elsewhere in the body (*Yu et al., 2023*).

Additionally, three neuron classes (IL2, HOB, and R5B) express *tbh-1* but lack expression of any other genes in canonical monoaminergic pathways, including *bas-1* (*Table 2*). Taken together, canonical monoaminergic pathway genes are expressed in unconventional combinations in several neuron classes, pointing toward the existence of yet undiscovered amino acid-derived neuronal signaling systems.

## Neurons devoid of canonical neurotransmitter pathway genes may define neuropeptide-only neurons

We identified neurons that do not express any conventional, well-characterized vesicular neurotransmitter transporter families, namely UNC-17/VAChT, CAT-1/VMAT (the only SLC18 family members), UNC-47/VGAT (only SLC32 family member), or EAT-4/VGLUT (an SLC17 family member). Six sex-shared neurons (AVH, BDU, PVM, PVQ, PVW, RMG) and one male-specific neuron (SPD) fall into this category. Most of these neurons exhibit features that are consistent with them being entirely neuropeptidergic. First, electron microscopy has revealed a relative paucity of clear synaptic vesicles in most of these neurons (*White et al., 1986*; *Cook et al., 2019*; *Witvliet et al., 2021*). Second, not only do these neurons express a multitude of neuropeptide-encoding genes (*Taylor et al., 2021*), but they also display a dense interconnectivity in the 'wireless' neuropeptidergic connectome (*Ripoll-Sánchez et al., 2023*).

That said, electron microscopy shows that some of the neurons devoid of conventional neurotransmitter pathway genes generate synapses with small, clear synaptic vesicles, indicative of the use of non-peptidergic transmitters (e.g. the sex-shared RMG and PVM neurons or the male-specific SPD neurons) (*White et al., 1986*; *Cook et al., 2019*; *Witvliet et al., 2021*). It is therefore conceivable that either conventional neurotransmitters utilize non-conventional neurotransmitter synthesis and/ or release pathways, or that completely novel neurotransmitter systems remain to be discovered. Although the *C. elegans* genome does not encode additional members of the SLC18A2/3 (*cat-1/VMAT, unc-17/VAChT*) or SLC32A1 (*unc-47/VGAT*) family of vesicular neurotransmitter transporters, it does contain a number of additional members of the SLC17A6/7/8 (VGLUT) family (*Hobert, 2013*).

These may serve as non-canonical vesicular transporters of more uncommon neurotransmitters or, alternatively, may be involved in modulating release of glutamate (*Serrano-Saiz et al., 2020*; *Choi et al., 2021*). Uncharacterized paralogs of bona fide neurotransmitter uptake transporters (SLC6 superfamily) may also have functions in neurotransmitter release rather than uptake. However, based on CeNGEN scRNA data, no robust or selective expression of these SLC17 or SLC6 family members is observed in these 'orphan neurons'.

## Co-transmission of multiple neurotransmitters

Our analysis expands the repertoire of neurons that co-transmit multiple neurotransmitters (*Figure 3*). Neurotransmitter co-transmission has been observed in multiple combinations in the vertebrate brain (*Wallace and Sabatini, 2023*). In *C. elegans,* the most frequent co-transmission configurations are a classic, fast transmitter (acetylcholine or glutamate) with a monoamine. Co-transmission of two distinct monoaminergic systems also exists. In several cases, however, it is not clear whether the second neurotransmitter is indeed used for communication or whether its presence is merely a reflection of this neuron being solely a clearance neuron. For example, the glutamatergic AIM neuron stains positive for serotonin, which it uptakes via the uptake transporter MOD-5, but it does not express the vesicular monoamine transporter *cat-1/VMAT* (*Figures 3, 6, and 8*, *Tables 1 and 2*).

Co-transmission of small, fast-acting neurotransmitters (glutamate, GABA, acetylcholine) does exist, but it is rare (*Figure 3*). The most prominent co-transmission configuration is acetylcholine with glutamate, but acetylcholine can also be co-transmitted with GABA. There are no examples of co-transmission of glutamate and GABA, as observed in several regions of the vertebrate brain (*Wallace and Sabatini, 2023*). There are also examples of possible co-transmission of three transmitters (*Figure 3*).

Interestingly, co-transmission appears to be much more prevalent in the male-specific nervous system, compared to the sex-shared nervous system (*Figure 3*, *Supplementary file 3*). This may relate to male-specific neurons displaying a greater degree of anatomical complexity compared to the hermaphrodite nervous system, both in terms of branching patterns and extent of synaptic connectivity (*Jarrell et al., 2012*; *Cook et al., 2019*). Given that all co-transmitting neurons display multiple synaptic outputs (*Cook et al., 2019*), it appears possible that each individual neurotransmitter secretory system is distributed to distinct synapses. Based on vertebrate precedent (*Wallace and Sabatini, 2023*), co-release from the same vesicles is also possible.

## Sexual dimorphisms in neurotransmitter usage

The observation of sexual dimorphisms in neurotransmitter abundance in specific regions of the mammalian brain has been one of the earliest molecular descriptors of neuronal sex differences in mammals (*McCarthy et al., 1997*). However, it has remained unclear whether such differences are the result of the presence of sex-specific neurons or are indications of distinctive neurotransmitter usage in sex-shared neurons. Using *C. elegans* as a model, we have been able to precisely investigate (a) whether sex-specific neurons display a bias in neurotransmitter usage and (b) whether there are neurotransmitter dimorphisms in sex-shared neurons (*Pereira et al., 2015*; *Gendrel et al., 2016*; *Serrano-Saiz et al., 2017b*) (this paper). We found that male-specific neurons display a roughly similar proportional usage of individual neurotransmitter systems and note that male-specific neurons display substantially more evidence of co-transmission, a possible reflection of their more elaborate morphology and connectivity. We also confirmed evidence for sexual dimorphisms in neurotransmitter usage in sex-shared neurons (*Supplementary file 4*), which are usually correlated with sexual dimorphisms in synaptic connectivity of these sex-shared neurons (*Cook et al., 2019*).

## Neurotransmitter pathway genes in glia and gonad

Neurotransmitter uptake is a classic function of glial cells across animal phylogeny (*Henn and Hamberger, 1971*), and such uptake mechanisms are observed in *C. elegans* as well. Previous reports demonstrated glutamate uptake by CEPsh (*Katz et al., 2019*) and GABA uptake by GLR glia (*Gendrel et al., 2016*). We now add to this list betaine uptake by most glia, as inferred from the expression pattern of SNF-3/BGT1 (*Figure 9*, *Supplementary file 1*).

Studies in vertebrates have also suggested that specific glial cell types synthesize and release several neurotransmitters (*Araque et al., 2014*; *Savtchouk and Volterra, 2018*). For example,

astrocytes were recently shown to express VGLUT1 to release glutamate (*de Ceglia et al., 2023*). Evidence of neurotransmitter synthesis and release also exists in *C. elegans* glia. Previous work indicated that glia associated with male-specific spicule neurons synthesize (through *cat-2/TH* and *bas-1/AAAD*) the monoaminergic transmitter dopamine to control sperm ejaculation (*LeBoeuf et al., 2014*). Our identification of *cat-1/VMAT* expression in these glia indicate that dopamine is released via the canonical vesicular monoamine transporter. We also detected expression of *bas-1/AAAD* in additional male and hermaphrodite glia, indicating the production of other signaling substances released by these glia. *bas-1* has indeed recently been shown to be involved in the synthesis of a class of unconventional serotonin derivatives (*Yu et al., 2023*).

There have been previous reports on GABA synthesis and release from the AMsh glial cell type (*Duan et al., 2020*; *Fernandez-Abascal et al., 2022*). We were not able to detect AMsh with anti-GABA staining, nor with reporter alleles of *unc-25/GAD*. However, since very low levels of *unc-25* are observed in the AMsh scRNA datasets (*Taylor et al., 2021*; *Purice et al., 2023*), the abundance of GABA in AMsh may lie below conventional detection levels.

Outside the nervous system, the most prominent and functionally best characterized usage of neurotransmitters lies in the hermaphrodite somatic gonad, which has been shown to synthesize octopamine and use it to control oocyte quiescence (*Alkema et al., 2005*; *Kim et al., 2021*). Intriguingly, we also detected *tbh-1, tdc-1,* and *cat-1* expression in the somatic gonad of the male, specifically the vas deferens, which is known to contain secretory granules that are positive for secretory molecular markers (*Nonet et al., 1993*). The presence of octopamine is unexpected because, unlike oocytes, sperm are not presently known to require monoaminergic signals for any aspect of their maturation. It will be interesting to assess sperm differentiation and function of *tbh-1* or *tdc-1* mutant animals. The usage of monoaminergic signaling systems in the gonad is not restricted to *C. elegans* and has been discussed in the context of sperm functionality and oocyte maturation in vertebrates (*Mayerhofer et al., 1999*; *Ramírez-Reveco et al., 2017*; *Alhajeri et al., 2022*).

## Comparing approaches and caveats of expression pattern analysis

Our analysis also provides an unprecedented and systematic comparison of antibody staining, scRNA transcript data, reporter transgene expression, and knock-in reporter allele expression. The bottom-line conclusions of these comparisons are: (1) Reporter alleles reveal more sites of expression than fosmid-based reporters. It is unclear whether this is due to the lack of *cis*-regulatory elements in fosmid-based reporters or issues associated with the multicopy nature of these reporters (e.g. RNAi-based gene silencing of multicopy arrays or squelching of regulatory factors). Another factor to consider is that neuron identification for most fosmid-based reporters was carried out prior to the introduction of NeuroPAL. Consequently, errors occasionally occurred, as exemplified by the misidentification of neuron IDs for CA7 and CP7 in previous instances (*Serrano-Saiz et al., 2017b*). (2) The best possible reporter approaches (i.e. reporter alleles) show very good overlap with scRNA data, thereby validating each approach. However, our comparisons also show that no single approach is perfect. CeNGEN scRNA data can miss transcripts and can also show transcripts in cells in which there is no independent evidence for gene or protein expression. Conversely, antibody staining displays vagaries related to staining protocols and protein localization, which can be overcome with reporter approaches, but the price to pay with reporter alleles is that if they are based on SL2 or T2A strategies, they may fail to detect additional levels of posttranslational regulation, which may result in protein absence even in the presence of transcripts. The existence of such mechanisms may be a possible explanation for cases where the expression of synthesis and/or transport machinery expression does not match up (e.g. *tdc-1*(-); *tbh-1*(+) neurons).

Our detailed analysis of reporter allele expression has uncovered several cases where expression of a neurotransmitter pathway gene in a given neuron class appears very low and variable from animal to animal. Such variability only exists when expression is dim, thus one possible explanation for it is that expression levels merely hover around an arbitrary microscopical detection limit. However, we cannot rule out the other possibility that this may also reflect true on/off variability of gene expression. Taking this notion a step further, we cannot exclude the possibility that expression observed with reporter alleles misses sites of expression. This possibility is raised by our inability to detect *unc-25/GAD* reporter allele expression in AMsh glia (*Duan et al., 2020*; *Fernandez-Abascal et al., 2022*) or *eat-4* reporter allele expression in AVL and DVB neurons, in which some (but not other) multicopy

reporter transgenes revealed expression of the respective genes (*Li et al., 2023*). Functions of these genes in the respective cell types were corroborated by cell-type-specific RNAi experiments and/or rescue experiments; whether there is indeed very low expression of these genes in those respective cells or whether drivers used in these studies for knock-down and/or rescue produce very low expression in other functionally relevant cells remains to be resolved.

## Conclusions

In conclusion, we have presented here the most complete neurotransmitter map that currently exists for any animal nervous system. Efforts to map neurotransmitter usage on a system-wide level are underway in other organisms, most notably, *Drosophila melanogaster* (*Deng et al., 2019*; *Eckstein et al., 2024*). The *C. elegans* neurotransmitter map presented here comprises a critical step toward deciphering information flow in the nervous system and provides valuable tools for studying the genetic mechanisms underlying cell identity specification. Moreover, this neurotransmitter map opens new opportunities for investigating sex-specific neuronal differentiation processes, particularly in the male-specific nervous system, where a scarcity of molecular markers has limited the analysis of neuronal identity control. Lastly, our analysis strongly suggests that additional neurotransmitter systems remain to be identified.

While the gene expression patterns delineated here enable informed predictions about novel neuronal functions and neurotransmitter identities, further investigations involving genetic perturbations, high-resolution imaging, complementary functional assays, and analyses across developmental stages are needed to shed further light on neurotransmitter usage. Nonetheless, this comprehensive neurotransmitter map provides a robust foundation for deciphering neural information flow, elucidating developmental mechanisms governing neuronal specification, exploring sexual dimorphisms in neuronal differentiation, and potentially uncovering novel neurotransmitter systems awaiting characterization.

# Materials and methods
## Transgenic reporter strains

Knock-in reporter alleles were generated either by SunyBiotech (*syb* alleles) or in-house (*ot* alleles) using CRISPR/Cas9 genome engineering. Most genes were tagged with a nuclear-targeted *gfp* sequence (*gfp* fused to *his-44*, a histone *h2b* gene) at the 3' end of the locus to capture all isoforms, except *tdc-1* which was tagged at the 5' end. For *unc-25*, both isoforms were individually tagged since a single tag would not capture both. Transgene schematics are shown in *Figure 2*.

Reporter alleles generated in this study:

> *unc-25(ot1372[unc-25a.1c.1::t2a:gfp::h2b]) III*
> *unc-25(ot1536[unc-25b.1::t2a::gfp::h2b]) III*
> *unc-46(syb7278[unc-46::sl2::gfp:h2b]) V*
> *unc-47(syb7566[unc-47::sl2::gfp::h2b]) III*
> *cat-1(syb6486[cat-1::sl2::gfp::h2b]) X*
> *tph-1(syb6451[tph-1::sl2::gfp::h2b]) II*
> *tbh-1(syb7786[tbh-1::sl2::gfp::h2b]) X*
> *tdc-1(syb7768[gfp::linker::h2b::t2a::tdc-1]) II*
> *cat-2(syb8255[cat-2::sl2::gfp::h2b]) II*
> *snf-3(syb7290[snf-3::TagRFP::sl2::gfp::h2b]) II*
> *oct-1(syb8870[oct-1::sl2::gfp::h2b]) I*
> *hdl-1(syb1048[hdl-1::gfp]) IV*
> *hdl-1(syb4208[hdl-1::t2a::3xnls::cre]) IV*

Since we did not detect fluorophore signals in the *hdl-1(syb1048[hdl-1::gfp])* strain, we attempted to amplify low-level signals, by inserting Cre recombinase at the C-terminus of the *hdl-1* locus (*hdl-1(syb4208[hdl-1::t2a::3xnls::cre])*). We crossed this strain to the recently published 'Flexon' strain (*arTi361[rps-27p::gfp"flexon"-h2b::unc-54–3'UTR]*) (*Shaffer and Greenwald, 2022*). Even low expression of *hdl-1* should have led to Cre-mediated excision of the flexon stop cassette, which is designed to abrogate gene expression by a translational stop and frameshift mutation, and subsequently can

result in strong and sustained *gfp* expression under the control of the *rps-27* promoter and thereby providing information about cell-specific *hdl-1* expression. However, no robust, consistent reporter expression was seen in *hdl-1(syb4208[hdl-1::t2a::3xnls::cre]); arTi361[rps-27p::gfp"flexon"-h2b::unc-54–3'UTR]* animals.

Three of the reporter alleles that we generated were already previously examined in specific cellular contexts:

*unc-17(syb4491[unc-17::t2a::gfp:h2b]) IV* (*Vidal et al., 2022*)
*eat-4(syb4257[eat-4::t2a::gfp::h2b]) III* (*Vidal et al., 2022*)
*bas-1(syb5923[bas-1::sl2::gfp::h2b]) III* (*Yu et al., 2023*)

One of the reporter alleles was obtained from the Caenorhabditis Genetics Center (CGC):

*mod-5(vlc47[mod-5::t2a::mNeonGreen]) I* (*Maicas et al., 2021*)

## Microscopy and image processing

For adult animal imaging, 15–25 (exact number depending on the difficulty of neuron ID) same-sex L4 worms were grouped on NGM plates 6–9 hr prior to imaging to control for accurate staging and avoid mating. Young adult worms were then anesthetized using 50–100 mM sodium azide and mounted on 5% agarose pads on glass slides. Z-stack images were acquired with ZEN software using Zeiss confocal microscopes LSM880 and LSM980 or a Zeiss Axio Imager Z2 and processed with ZEN software or FIJI (*Schindelin et al., 2012*) to create orthogonal projections. Brightness and contrast, and in some cases gamma values, were adjusted to illustrate dim expression and facilitate neuron identification.

## Neuron class and cell-type identification

Neuron classes were identified by crossing the *gfp* reporter alleles with the landmark strain 'NeuroPAL' (allele *otIs669* or *otIs696*, for bright reporters and dim reporters, respectively) and following published protocols (*Tekieli et al., 2021*; *Yemini et al., 2021*) (also see 'lab resources' at hobertlab.org). For neuron identification of the *eat-4(syb4257)*, *unc-46(syb7278)*, and *unc-47(syb7566)* reporter alleles in hermaphrodites, the reporter alleles were also crossed into the fosmid-based reporter transgenes of the same gene [*eat-4(otIs518)*, *unc-46(otIs568)*, *and unc-47(otIs564)*] as a 'first-pass' to identify potential non-overlapping expression of the two alleles. For *tph-1(syb6451)* analysis, an *eat-4* fosmid-based reporter (*otIs518*) was also used. For identification of VC4, VC5, HSN, and uv1, an *ida-1p::mCherry* integrant (LX2478, *lin-15(n765ts) X; vsIs269[ida-1::mCherry]*) was also used in some cases (*Fernandez et al., 2020*). For phasmid neurons, dye-filling with DiD (Thermo Fisher Scientific) was sometimes used to confirm neuron ID. For glial expression, a panglial reporter *otIs870[mir-228p::3xnls::TagRFP]* was used. For hypodermal cells identification, a *dpy-7p::mCherry* reporter *stIs10166 [dpy-7p::his-24::mCherry+unc-119(+)]* was used (*Liu et al., 2009*).

## Resource availability

### Lead contact

Oliver Hobert (or38@columbia.edu) is the Lead Contact.

### Materials availability

All newly generated strains are available at the Caenorhabditis Genetics Center (CGC).

## Acknowledgements

We thank Chi Chen for generating nematode strains. We thank Emily Bayer, James Rand, Piali Sengupta, and Esther Serrano-Saiz for comments on the manuscript, Frank Schroeder and Marie Gendrel for discussion and communicating unpublished results, Aakanksha Singhvi for discussing glia scRNA data and Michael Koelle for an *ida-1* reporter strain. Some strains were provided by the CGC, which is funded by NIH Office of Research Infrastructure Programs (P40 OD010440). This work was funded by the Howard Hughes Medical Institute and by NIH R01 NS039996.

## Additional information

### Funding

| Funder | Grant reference number | Author |
| --- | --- | --- |
| National Institutes of Health | NS039996 | Oliver Hobert |
| Howard Hughes Medical Institute | | Oliver Hobert |
| National Institutes of Health | Office of Research Infrastructure Programs P40 OD010440 | Oliver Hobert |

The funders had no role in study design, data collection and interpretation, or the decision to submit the work for publication.

### Author contributions

Chen Wang, Conceptualization, Data curation, Formal analysis, Investigation, Visualization, Writing – original draft; Berta Vidal, Surojit Sural, Curtis Loer, G Robert Aguilar, Daniel M Merritt, Itai Antoine Toker, Merly C Vogt, Cyril C Cros, Investigation, Visualization, Writing – review and editing; Oliver Hobert, Conceptualization, Supervision, Funding acquisition, Writing – original draft, Project administration

### Author ORCIDs

Chen Wang ![iD] https://orcid.org/0000-0002-3363-139X
Surojit Sural ![iD] https://orcid.org/0000-0002-0422-9799
G Robert Aguilar ![iD] https://orcid.org/0000-0001-6926-0319
Itai Antoine Toker ![iD] https://orcid.org/0000-0002-0349-1808
Cyril C Cros ![iD] https://orcid.org/0000-0002-8812-1194
Oliver Hobert ![iD] https://orcid.org/0000-0002-7634-2854

Reviewer #1 (Public review): https://doi.org/10.7554/eLife.95402.3.sa1
Reviewer #2 (Public review): https://doi.org/10.7554/eLife.95402.3.sa2
Reviewer #3 (Public review): https://doi.org/10.7554/eLife.95402.3.sa3
Author response https://doi.org/10.7554/eLife.95402.3.sa4

## Additional files

### Supplementary files

• Supplementary file 1. Single-cell RNA (scRNA) data for neurotransmitters in the hermaphrodite. Here, we show expression of previous reporters and reporter alleles used in this study, compared to scRNA data. Note that scRNA expression values for *eat-4* and *unc-47* can be unreliable because they were overexpressed to isolate individual neurons for scRNA analysis (*Taylor et al., 2021*).

• Supplementary file 2. Updated expression patterns of neurotransmitter pathway genes in hermaphrodites.

• Supplementary file 3. Updated expression patterns of neurotransmitter pathway genes in male-specific neurons.

• Supplementary file 4. Summary of sexually dimorphic use of neurotransmitter pathway genes in sex-shared neurons.

• Supplementary file 5. Summary of updates to expression patterns of classic neurotransmitter pathway genes.

• MDAR checklist

### Data availability

All data generated or analysed during this study are included in the manuscript and supporting files.

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
