## [Editor Report · eLife assessment]

This **fundamental** study reports the most comprehensive neurotransmitter atlas of any organism to date, using fluorescent knock-in reporter lines. The work is comprehensive, rigorous, and **compelling**. The tool will be used by broad audience of scientists interested in neuronal cell type differentiation and function, and could be a seminal reference in the field.

---

## [Referee Report · Reviewer #1 (Public review)]

Summary:

Wang and colleagues conducted a study to determine the neurotransmitter identity of all neurons in *C. elegans* hermaphrodites and males. They used CRISPR technology to introduce fluorescent gene expression reporters into the genomic loci of NT pathway genes. This approach is expected to better reflect in vivo gene expression compared to other methods like promoter- or fosmid-based transgenes, or available scRNA datasets. The study presents several noteworthy findings, including sexual dimorphisms, patterns of NT co-transmission, neuronal classes that likely use NTs without direct synthesis, and potential identification of unconventional NTs (e.g. betaine releasing neurons). The data is well-described and critically discussed, including a comparison with alternative methods. Although many of the observations and proposals have been previously discussed by the Hobert lab, the current study is particularly valuable due to its comprehensiveness. This NT atlas is the most complete and comprehensive of any nervous system that I am aware of, making it an extremely important tool for the community.

Strengths:

Very compelling study presenting the most comprehensive neurotransmitter (NT) map of any model so far, using state-of-the art tools and validations. The work is very important not only as a resource but also for our understanding that (NT) function of neurons is best understood taking into consideration the full set of genes implicated in NT metabolism and transport.

Weaknesses:

None, all have been addressed.

---

## [Referee Report · Reviewer #2 (Public review)]

Summary:

Together with the known anatomical connectivity, molecular atlasses paves the way toward functional maps of the nervous system of *C. elegans*. Along with the analysis of previous scRNA sequencing and reporter strains, new expression patterns are generated for hermaphrodite and males based on CRISPR-knocked-in GFP reporter strains and the use of the color-coded Neuropal strain to accurately identify neurons. Beyond a map of the known neurotransmitters (GABA, Acetylcholine, Glutamate, dopamine, serotonin, tyramine, octopamine), the atlas also identifies neurons likely using betaine and suggests sets of neurons employing new unknown monoaminergic transmission, or using exclusively peptidergic neurotransmission.

Strengths:

The use of CRISPR reporter alleles and of the Neuropal strain to assign neurotransmitter usage to each neuron is much more rigourous than previous analysis and reveal intriguing differences between scRNA seq, fosmid reporter and CRISPR knock-in approaches. The differences between approaches are discussed.

Weaknesses:

All have been addressed.

---

## [Referee Report · Reviewer #3 (Public review)]

Summary:

In this paper, Wang et al. provides the most comprehensive description and comparison of the expression of the different genes required to synthesize, transport and recycle the most common neurotransmitters (Glutamate, Acetylcholine, GABA, Serotonin, Dopamine, Octopamine and Tyramine) used by hermaphrodite and male *C. elegans*. This paper will be a seminal reference in the field. Building and contrasting observations from previous studies using fosmid, multicopy reporters and single cell sequencing, they now describe CRISPR/Cas-9-engineered reporter strains that, in combination with the multicolor pan-neuronal labeling of all *C. elegans* neurons (NeuroPAL), allows rigorous elucidation of neurotransmitter expression patterns. These novel reporters also illuminate previously unappreciated aspects of neurotransmitter biology in *C. elegans*, including sexual dimorphism of expression patterns, co-transmission and the elucidation of cell-specific pathways that might represent new forms of neurotransmission.

Strengths:

The authors set to establish neurotransmitter identities in *C. elegans* males and hermaphrodites via varying techniques, including integration of previous studies, examination of expression patterns and generation of endogenous CRISPR-labeled alleles. Their study is comprehensive, detailed and rigorous, and achieve the aims. It is an excellent reference for the field, particularly those interested in biosynthetic pathways of neurotransmission and their distribution in vivo, in neuronal and non-neuronal cells.

Weaknesses:

No weaknesses noted. The authors do a great job linking their characterizations with other studies and techniques, leading credence to their findings. As the authors note, there are sexually dimorphic differences across animals, and varying expression patterns of enzymes. While it is unlikely there will be huge differences in the reported patterns across individual animals, it is possible that these expression patterns could vary developmentally, or based on physiological or environmental conditions.

---

## [Author Response]

The following is the authors’ response to the original reviews.

We would like to thank the reviewers and editor for their helpful comments and suggestions. In response, we have revised the manuscript in two main ways:

(1) To address the comments about rearranging figures and tables, we added a new Figure 3 that summarizes neurotransmitter assignments across all neuron classes. Our rationale for this change is detailed below.

(2) To address the comment on clarifying neurotransmitter synthesis versus uptake, we analyzed two additional reporter alleles that tag the monoamine uptake transporters for 5-HT and potentially tyramine. These results are now presented in a new Figure 8 and corresponding sections in the manuscript. Related tables have been updated to include this expression data. Two more authors have been added due to their contributions to these experiments.

For more detailed changes, please see our responses to the specific reviewer's comments as well as the revised manuscript.

**Public Reviews:**

**Reviewer #1 (Public Review):**
Wang and colleagues conducted a study to determine the neurotransmitter identity of all neurons in *C. elegans* hermaphrodites and males. They used CRISPR technology to introduce fluorescent gene expression reporters into the genomic loci of NT pathway genes. This approach is expected to better reflect in vivo gene expression compared to other methods like promoter- or fosmid-based transgenes, or available scRNA datasets. The study presents several noteworthy findings, including sexual dimorphisms, patterns of NT co-transmission, neuronal classes that likely use NTs without direct synthesis, and potential identification of unconventional NTs (e.g. betaine releasing neurons). The data is well-described and critically discussed, including a comparison with alternative methods. Although many of the observations and proposals have been previously discussed by the Hobert lab, the current study is particularly valuable due to its comprehensiveness. This NT atlas is the most complete and comprehensive of any nervous system that I am aware of, making it an extremely useful tool for the community.
**Reviewer #2 (Public Review):**
Summary:Together with the known anatomical connectivity of *C. elegans*, a neurotransmitter atlas paves the way toward a functional connectivity map. This study refines the expression patterns of key genes for neurotransmission by analyzing the expression patterns from CRISPR-knocked-in GFP reporter strains using the color-coded Neuropal strain to identify neurons. Along with data from previous scRNA sequencing and other reporter strains, examining these expression patterns enhances our understanding of neurotransmitter identity for each neuron in hermaphrodites and the male nervous system. Beyond the known neurotransmitters (GABA, Acetylcholine, Glutamate, dopamine, serotonin, tyramine, octopamine), the atlas also identifies neurons likely using betaine and suggests sets of neurons employing new unknown monoaminergic transmission, or using exclusively peptidergic transmission.Strengths:The use of CRISPR reporter alleles and of the Neuropal strain to assign neurotransmitter usage to each neuron is much more rigorous than previous analysis and reveals intriguing differences between scRNA seq, fosmid reporter, and CRISPR knock-in approaches. Among other mechanisms, these differences between approaches could be attributed to 3'UTR regulatory mechanisms for scRNA vs. knockin or titration of rate-limited negative regulatory mechanisms for fosmid vs. knockin. It would be interesting to discuss this and highlight the occurrences of these potential phenomena for future studies.

We recognize that readers of this study may be interested in understanding the differences between the three approaches. Therefore, in the Introduction, we addressed the potential risk of overexpression artifacts associated with multicopy transgenes, such as fosmid-based reporters, which can affect rate-limiting negative regulatory mechanisms. Additionally, in the Discussion, we included a section titled 'Comparing approaches and caveats of expression pattern analysis' to further explore these comparative methods and their associated nuances.

Weaknesses:For GABAergic transmission, one shortcoming arises from the lack of improved expression pattern by a knockin reporter strain for the GABA recapture symporter snf-11. In its absence, it is difficult to make a final conclusion on GABA recapture vs GABA clearance for all neurons expressing the vesicular GABA transporter neurons (unc-47+) but not expressing the GAD/UNC-25 gene e.g. SIA or R2A neurons. At minima, a comparison of the scRNA seq predictions versus the snf-11 fosmid reporter strain expression pattern would help to better judge the proposed role of each neuron in GABA clearance or recycling.

The *snf-11* fosmid-based reporter data shows very good overlap with scRNA seq predictions (now included in Supp. Table S1).

But there are two much stronger reasons why we did not seek to further the analysis of expression of the *snf-11* GABA uptaker:

(1) Due to available anti-GABA staining data, we do know which neurons have the potential to take up GABA (via SNF-11).

(2) Focusing on SNF-11 *function* rather than *expression*, we can ask which neurons lose anti-GABA staining in *snf-11* mutants.

Both of these types of analyses have been done in an earlier study from our lab (Gendrel et al., 2016, PMID 27740909), which, among other things, investigated GABA uptake mechanisms via SNF-11. Apart from analyzing the expression of a fosmid-based *snf-11* reporter, we immunostained worms for GABA in both *snf-11* mutant and wild type backgrounds (results summarized in Tables 1 and 2 of Gendrel et al.). Of the neurons that typically stain for GABA (Table 1, Gendrel et al.), two neuron classes (ALA and AVF) lost the staining in *snf-11* mutants, suggesting that these neurons likely uptake GABA via SNF-11. Importantly, one of the neurons the reviewer mentioned, R2A, stains for GABA in both wild type and *snf-11* mutants, indicating that it likely does not uptake GABA via SNF-11. The other neuron mentioned, SIA, does not stain for GABA in wild type (Table 2, Gendrel et al.), hence not a GABA uptake neuron. In cases like SIA and other neurons, where a neuron does not express *unc-25* but does express *unc-47* reporters (either fosmid or CRISPR reporter alleles), we speculate that UNC-47 transport another neurotransmitter.

Considering the complexities of different tagging approaches, like T2A-GFP and SL2-GFP cassettes, in capturing post-translational and 3'UTR regulation is important. The current formulation is simplistic. e.g. after SL2 trans-splicing the GFP RNA lacks the 5' regulatory elements, T2A-GFP self-cleavage has its own issues, and the his-44-GFP reporter protein does certainly have a different post-translational life than vesicular transporters or cytoplasmic enzymes.

Yes, agreed, these points are mentioned in the Introduction and discussed in "Comparing approaches and caveats of expression pattern analysis" in the Discussion.

Do all splicing variants of neurotransmitter-related genes translate into functional proteins? The possibility that some neurons express a non-functional splice variant, leading to his-74-GFP reporter expression without functional neurotransmitter-related protein production is not addressed.

We thank the reviewer for bringing up this really interesting point, which we had not considered. First and foremost, with the exception of *unc-25* (discussed in the next point), for all other genes that produce multiple splice forms, we made sure to append our tag (at 5’ or 3’ end) such that the expression of all splice forms is captured. The reviewer raises the interesting point that in an alternative splicing scenario, some of the cells that express the primary transcript may “switch” to an inactive form. While we cannot exclude this possibility, we have confirmed by sequence analysis in WormBase that in five of the six cases where there is alternative splicing, the alternatively spliced exon lies outside the conserved, functionally relevant (enzymatic or structural) domain. In one case, *unc-25*, a shorter isoform is produced that does cut into the functionally relevant domain; however, since all *unc-25* reporter allele expression cells are also staining positive for GABA, this may not be an issue.

Also, one tagged splice variant of unc-25 is expected to fail to produce a GFP reporter, can this cause trouble?

Yes, there is indeed a third splice variant of *unc-25* with an alternative C-terminus. To address potential expression of this isoform, we CRISPR-engineered another reporter, *unc-25(ot1536[unc-25b.1::t2a::gfp::h2b])*, in which the inserted *t2a::gfp::h2b* sequences are fused to the C-terminus of the alternative splice form, but we did not observe any expression of this reporter. Now included in the manuscript.

**Reviewer #3 (Public Review):**
Summary:In this paper, Wang et al. provide the most comprehensive description and comparison of the expression of the different genes required to synthesize, transport, and recycle the most common neurotransmitters (Glutamate, Acetylcholine, GABA, Serotonin, Dopamine, Octopamine, and Tyramine) used by hermaphrodite and male *C. elegans*. This paper will be a seminal reference in the field. Building and contrasting observations from previous studies using fosmid, multicopy reporters, and single-cell sequencing, they now describe CRISPR/Cas-9-engineered reporter strains that, in combination with the multicolor pan-neuronal labeling of all *C. elegans* neurons (NeuroPAL), allows rigorous elucidation of neurotransmitter expression patterns. These novel reporters also illuminate previously unappreciated aspects of neurotransmitter biology in *C. elegans*, including sexual dimorphism of expression patterns, cotransmission, and the elucidation of cell-specific pathways that might represent new forms of neurotransmission.Strengths:The authors set out to establish neurotransmitter identities in *C. elegans* males and hermaphrodites via varying techniques, including integration of previous studies, examination of expression patterns, and generation of endogenous CRISPR-labeled alleles. Their study is comprehensive, detailed, and rigorous, and achieves the aims. It is an excellent reference for the field, particularly those interested in biosynthetic pathways of neurotransmission and their distribution in vivo, in neuronal and non-neuronal cells.Weaknesses:No weaknesses were noted. The authors do a great job linking their characterizations with other studies and techniques, giving credence to their findings. As the authors note, there are sexually dimorphic differences across animals and varying expression patterns of enzymes. While it is unlikely there will be huge differences in the reported patterns across individual animals, it is possible that these expression patterns could vary developmentally, or based on physiological or environmental conditions. It is unclear from the study how many animals were imaged for each condition, and if the authors noted changes across individuals during development (could be further acknowledged in the discussion?)

We have updated the Methods section to specify the number of animals used for imaging. We agree with the reviewer that documenting the developmental dynamics of neurotransmitter expression would be interesting. However, except for one gene (*tph-1*, Fig. S2), we did not analyze the expression during different developmental stages for most genes in this study. Following the reviewer's suggestion, we have included this as a potential future direction in "Conclusions" at the end of the revised manuscript.

**Recommendations for the authors**:After the consultation session, a common suggestion from the reviewers is to bring the tables more upfront, perhaps even in the form of legible main Figures and in alphabetical order of neurons; since we believe that the study will be in the long-term often used for these data; while the Figures with fluorescent expression patterns could be moved to the supplemental information.

We appreciate the reviewers' and editor's acknowledgment of the tables' possibly frequent usage by the field. We have considered carefully how to order the data presentation. We prefer to keep most of the fluorescent figures in the main text because they convey important subtleties that we want the reader to be aware of.

To address the suggestions to bring key data more upfront, we have added an entirely new figure (Figure 3) before the ensuing data figures that summarized expression patterns of the fluorescent reporters. This new figure (A) summarizes the neurotransmitter use for all neuron classes and (B) illustrates this information within worm schematics, showing the position of neurons in the whole worm. This figure serves as a good overview of neurotransmitter assignments but also specifically refers to the more extensive data and supplementary tables with detailed notes. We believe this solution effectively balances the need for comprehensive information and ease of reference.

**Reviewer #1 (Recommendations for The Authors):**
Suggestions:(1) The study contains up to 10 Figures with gene expression patterns; however, I believe the community will use this paper mostly in the future for its summarizing tables. I wonder if it would be more useful to edit the tables and move them to the main figures while most fluorescent reporter images could be moved to the supplementary part.

Yes, as mentioned above, we made new summary table & schematic upfront. We do prefer to keep primary data in main figure body. Please see above (Public Review & Response).

(2) In the section titled 'Neurotransmitter Synthesis versus Uptake', the author's wording could be more careful. The data rather suggests functions for individual neuronal classes, such as clearance neurons or signaling neurons. However, these functions remain hypotheses until further detailed studies are conducted to test them.

These are fair points. We have made several improvements:

(1) In the referenced section, we added a sentence at the end of the paragraph on betaine to suggest the importance of future functional studies.

(2) We analyzed reporter allele expression for two additional genes: the known uptake transporter for 5-HT (*mod-5*, reporter allele *vlc47*) and the predicted uptake transporter for tyramine (*oct-1*, reporter allele *syb8870*). The results from these experiments are presented in the new Figure 8 and discussed in Results and Discussion correspondingly. We also collaborated with Curtis Loer, who conducted anti-5-HT staining in wild type and *mod-5* mutant animals (results shown in Figure 12). These experiments have enhanced our understanding of 5-HT uptake mechanisms and potential tyramine uptake mechanisms.

(3) At the end of the Conclusions, we emphasized the need for future detailed studies to test the functions of neurotransmitter synthesis and uptake.

(3) Page 21; add to the discussion: neurons could use mainly electrical synapses for communication. Especially for RMG neurons, this might be the case (in addition to neuropeptide communication).

“Main usage” is a difficult term to use. If there were neurons that are clearly devoid of any form of synaptic vesicle (small or DCV; note that RMG has plenty of DCVs), but show robust and reproducible electrical synapses, we would agree that such neurons could primarily be a “coupling” neuron. But this call is very hard to make for any *C. elegans* neuron (RMG included) and hence we prefer to not add further to an already quite long Discussion section.

(4) Page 23: I believe that multi-copy promoter-based transgenes (despite array suppression mechanisms) could be potentially more sensitive than single-copy insertion of fluorescent reporters. In our lab, we observed this a couple of times. This could be discussed.

We discuss this in "Comparing approaches and caveats of expression pattern analysis" in the Discussion.

We have also added a third possibility (i.e. technical issues related to neuron-ID) in the revised manuscript.

**Reviewer #2 (Recommendations For The Authors):**
Comment during consultation session: As for my feedback on the lack of an SNF-11 reporter strain, exercising more caution in their conclusions would suffice for me. Other comments are simple edits/discussion.

Please see above.

Several neurotransmitter symporters exist in the *C. elegans* genome, does any express specifically in the "orphan" UNC-47+ neurons?

Yes, good point, we considered this possibility, but of the >10 SLC6-family of neurotransmitter reporters, only the classic, de-orphanized ones that we discuss here in the paper show robust scRNA signals (as discussed in the paper) and none of those give clues about the orphan *unc-47*(+) neurons.

Based on UNC-47+ expression the article suggests a "Novel inhibitory neurotransmitter". Why would any new neurotransmitter using UNC-47 be necessarily inhibitory? The presence of one potential glycine-gated anion channel and one GPCR in *C. elegans* genome sounds poor evidence to suggest a sign of glycine or b-alanine transmission.

Yes, agreed, it does not need to be inhibitory. Fixed in Results and Discussion.

To help readers the expression of the knocked in GFP in neurons should not be reported as binary in table S1 which leads to a feeling of strong discrepancy between scRNA seq and CRISPR GFP, which is not the case.

There might be some misunderstanding regarding the coloring in this table. To clarify, the green-filled Excel cells denote the expression of reporters utilized in prior studies, rather than the CRISPR reporter alleles. Expression of the CRISPR alleles is instead indicated on the left side of the neuron names, marked as "CRISPR+" in green font. For signifying absence of expression, we used "no CRISPR" in red font in the first submission. We have now changed it into "CRISPR-" for greater clarity.

The variable expression of reporter GFP between individuals for the same neuron is intriguing. It is unclear if this is observed only for dim neurons or can be more of an ON/OFF expression.

Variability only occurs for dim expression. We have now clarified this point in Discussion, "Comparing approaches and caveats of expression pattern analysis".

The multiple occurrences of co-transmission, especially in male neurons, are interesting. It will be interesting in the future to establish whether the neurotransmitters are synaptically segregated or coreleased. As the section on sexual dimorphism of neurotransmitter usage does not discuss novel information coming from this study, it is not very necessary.

Agreed. We added this perspective to the Discussion, "Co-transmission of multiple neurotransmitters".

In the abstract, dopamine is missing in the main known transmitter.

Fixed. Thanks for spotting this.

**Reviewer #3 (Recommendations For The Authors):**
Great article. Minor suggestions to strengthen presentation:Figure 1B is hard to interpret. There could be more intuitive ways of representing the data and the methodologies that support a given expression pattern. Neurons should also be reordered by alphabetical order rather than expression levels to facilitate finding them.

We considered alternative ways of presenting this data, but, regrettably, did not come up with a better approach. To clarify, the primary focus of Fig. 1B is to compare expression of previously reported reporters and scRNA data, which was quite literally the initial impetus for our analysis, i.e. we noted strong scRNA signals that had not previously been supported by transgenic reporter data. For a comprehensive version of the table that includes more details on the expression of CRISPR reporter alleles, please refer to Table S1, which we referenced in the figure legend.

GFP-only channel images in Figures 3, 4, 5, and 9 sometimes show dim signals that the authors are highlighting as new findings. We recommend using the inverted grayscale version of that channel since the contrast of dim signals is more noticeable to the human eye rather than when the image is colorized.

Good point, we implemented these suggestions in the figures the reviewer mentioned, now re-numbered Figures 4, 5, 6, and 12. For Figure 6 (*tph-1*, *bas-1*, and *cat-1* expression in hermaphrodites), we used a new *cat-1* head image to reflect the newly identified ASI and AVL expression that wasn’t readily visible in the original projection used in the earlier version of this manuscript. We also added grayscale images in Figure 13 to reflect dim *tbh-1* expression in IL2 neurons more clearly.

A plan to integrate this new information into WormAtlas. The *C. elegans* community is characterized by the open sharing of information on platforms that are user-friendly and accessible. Ideally, the new information would not just 'erase' what was observed before but will describe the new observations and will let the community reach their own conclusions since there is no perfect method and even these CRISPR/Cas9 reporter strains are only proxy for gene expression that subject to post-transcriptional regulation since they depend on T2A and SL2 sequences.

We completely agree with the reviewer’s suggestion. We will coordinate with WormAtlas on integrating this new information.

In the case of neurons that were removed from using a specific neurotransmitter, like PVQ. What do the authors conclude overall, if it does not use glutamate, are there any new hypotheses to what it could be using?

Since all neurons express multiple neuropeptides, we hypothesize neurons such as PVQ may be primarily peptidergic. This is included in Discussion, "Neurons devoid of canonical neurotransmitter pathway genes may define neuropeptide-only neurons".

In Table S5, the I4 neuron is listed as a variable for eat-4 expression but in Table S1 it says that there was no CRISPR expression detected. Which one is correct?

Thanks for spotting this. Table S5 is correct, we saw very dim and variable expression of the *eat-4* reporter allele in I4. Table S1 is fixed now.

Additional discussion points that might be important for the community:CRIPSR strains used here should be deposited in the CGC.

Yes, all strains generated in this study have already been deposited to CGC.

It would be great to have an additional discussion point on how the neural clusters in CenGEN were defined based on the fosmid reporter expression, so in a way using the defining factor as one that was already defined by it might make results confusing.

Neural cluster definition in CeNGEN did not rely on isolated data points but on the combination of many expression reagents, each with its own shortcomings, but in combination providing reliable identification. Since one feedback we have gotten from many readers of our manuscript is that it is already very long as is, we prefer not to dilute the discussion further.

It would be important to discuss the rate of neurotransmitter genes that have variable expression patterns. Are any of those genes used in NeuroPAL to define specific neuronal classes? This is important to describe as NeuroPAL labeling is being used to define neuronal identity.

All the reporters used in NeuroPAL are promoter-based, very robust and do not include the full loci of genes, so they are not directly comparable with the CRISPR reporter alleles in this study. However, we recognize that some expression pattern variability could be confusing. We have discussed this more in the section "Comparing approaches and caveats of expression pattern analysis" in the Discussion.